# Structure Enables Effective Self-Localization of Errors in LLMs

**Ankur Samanta** [1 2]  **Akshayaa Magesh** [2]  **Ayush Jain** [2]  **Kavosh Asadi** [2]  **Youliang Yu** [2]  **Daniel R. Jiang** [2]
**Boris Vidolov** [2]  **Kaveh Hassani** [3]  **Paul Sajda** [1]  **Jalaj Bhandari** [* 2]  **Yonathan Efroni** [* 4]

## Abstract

Self-correction in language models remains elusive. In this work, we explore whether language models can explicitly localize errors in incorrect reasoning, as a path toward building AI systems that can effectively correct themselves. We introduce a prompting method that structures reasoning as discrete, semantically coherent thought steps, and show that models can localize errors more reliably within this structure than in conventional, unstructured chain-of-thought reasoning. Motivated by how the human brain monitors errors at discrete decision points and resamples alternatives, we introduce Iterative Correction Sampling of Thoughts (Thought-ICS), a self-correction framework. Thought-ICS iteratively prompts the model to generate reasoning one discrete and complete thought at a time—where each thought represents a deliberate decision by the model—creating natural boundaries for precise error localization. Upon verification, the model localizes the first erroneous step, and the system backtracks to generate alternative reasoning from the last correct point. When asked to correct reasoning verified as incorrect by an oracle, Thought-ICS achieves 20-40% self-correction lift. In a completely autonomous setting without external verification, it outperforms contemporary self-correction baselines.

## 1. Introduction

The ideal of self-correction in language models rests on a seemingly intuitive assumption: that repeatedly prompting a model to refine its output naturally converges to improved reasoning. In practice, this does not reliably happen. Observed gains often reflect brute-force resampling rather than deliberate self-correction (Huang et al., 2024; Tsui, 2025; Kumar et al., 2024). This can largely be attributed to failures in verification, confounding the models' true ability to localize and correct incorrect reasoning.

In this work, we first study self-correction in a setting with oracle verification (Section 5.1), sidestepping this bottleneck to isolate how effectively the model is able to explicitly correct its reasoning. Like humans, systems can only improve their reasoning to the extent they can verify it (Sutton, n.d.; Stechly et al., 2024; Kruger & Dunning, 1999). Prior work has shown that LLMs can self-correct when given the precise error location by an oracle localizer (Tyen et al., 2024), but this is perhaps unsurprising, as resampling from a correct prefix requires only the same abilities as initial generation. Our work aims to address this by asking: **can LLMs self-localize errors in their own reasoning?**

To study this, we need a framework that isolates the component abilities: verification (detecting errors), localization (identifying erroneous decision points), and correction (resampling an alternative path). For insight into how to structure such a system, we turn to neurological systems that have evolved to support error correction. In the brain, the anterior cingulate cortex monitors errors at the level of meaningful decision points, discrete actions or thoughts, neither individual neuron activations nor entire behavioral sequences (Botvinick et al., 2004; Carter et al., 1998; MacDonald et al., 2000). The brain then explores alternatives by selectively modifying decision points and simulating how reasoning would unfold from there (Miller & Cohen, 2001; Daw et al., 2005; Byrne, 2016; Gerstenberg, 2024). This suggests an architecture for self-correction: generate reasoning as discrete thoughts, localize errors at that granularity, and resample from the last correct step.

To instantiate this in language models, we introduce Iterative Correction Sampling of Thoughts (Thought-ICS). Standard chain-of-thought generates reasoning as a continuous token stream, difficult to retroactively parse into distinct steps. Thought-ICS instead operates in a Thought MDP: the model generates one thought at a time, making a deliberate decision at each step as to what the next semantically coherent reasoning step should be. This produces structured reasoning: a discrete chain of thoughts with principled boundaries, providing clarity during generation itself about where one

---

[1]Columbia University [2]Meta AI [3]Meta Superintelligence Labs [4]Tel Aviv University. Correspondence to: Ankur Samanta <as7416@columbia.edu>.

*Proceedings of the 43rd International Conference on Machine Learning*, Seoul, South Korea. PMLR 306, 2026. Copyright 2026 by the author(s).

step ends and another begins, enabling precise error localization and targeted intervention. An external scaffold enables backtracking: when an error is detected, alternative thought sequences are generated from the last verified-correct prefix. We show that this structure improves self-localization and self-correction over unstructured chain-of-thought, particularly for larger models.

Finally, we study self-correction in a setting with self-verification (Section 6), exploring how to design a fully autonomous system that deploys Thought-ICS as an inference time method. We identify the current limitations of verification without external ground truth, and show that existing self-correction systems can break more than they fix as a result. We demonstrate how to mitigate this, showing that Thought-ICS can indeed be used for self-guided correction in inference time.

We evaluate across eight models (3B–120B parameters), six reasoning benchmarks, and present the following contributions:

- We demonstrate that framing reasoning as a Thought MDP, where the model generates thought-by-thought, enables precise self-localization. Stronger models localize more precisely within this structure, yielding clean prefixes that support effective correction through backtracking and resampling.

- Self-correction within Thought-ICS with an oracle verifier consistently achieves 20–40% lift over chain-of-thought baselines, outperforming self-correction methods that lack explicit structure.

- We show that Thought-ICS deployed in a fully autonomous setting, without any oracle involvement, still achieves a net positive self-correction lift, outperforming contemporary self-correction methods.

## 2. Related Work

Among self-correction methods, two main approaches require neither external feedback nor correction-specific training: Self-Refine (Madaan et al., 2023) prompts the model to critique and regenerate its own response, while CoVe (Dhuliawala et al., 2023) derives verification questions from the chain-of-thought to guide revision. We compare our method, Thought-ICS, against both. Methods relying on external feedback include Reflexion (Shinn et al., 2023), which uses task outcome signals (e.g., test results) in agentic coding to generate verbal reflections stored in episodic memory, and RCI (Kim et al., 2023), which uses environment feedback for computer tasks in a recursive critique loop. As Huang et al. (2024); Stechly et al. (2024) show, without such external signals, models struggle to verify their own reasoning and often degrade performance through self-correction.

Tyen et al. (2024) decomposes self-correction into localization and correction, finding that LLMs cannot self-localize errors effectively, but can correct when given error locations by an oracle. We find that with appropriate structure, this limitation can be overcome. The aforementioned methods generate reasoning token-by-token and treat traces as monolithic, regenerating entirely based on holistic critique rather than localizing errors to specific steps. We instead generate thought-by-thought, enabling self-localization and targeted correction, fixing only what needs to be fixed. Prior work on thought-level generation (Yao et al., 2023; Hao et al., 2023; Feng et al., 2024; Wu et al., 2025) uses rule-based splitting and external scoring in tree search formulations, often producing segments that are not semantically coherent. We define a thought-MDP framework where the model determines its own thought boundaries, enabling self-localization, backtracking, and resampling for self-correction without external scoring. Iterative correction can be viewed as depth-first search in thought space: generate a trace, backtrack upon finding an error, and resample an alternative continuation. This contrasts with breadth-first approaches like beam search or tree search with value functions (Yao et al., 2023; Hao et al., 2023), which expand multiple candidates in parallel using learned or heuristic scoring. These approaches are complementary: tree search methods can improve initial trace quality, while our work tests the ability of the model to effectively backtrack and resample within its own reasoning. Prior work has also focused primarily on frontier models; we evaluate the emergence of self-corrective ability across model sizes, showing that given appropriate verification, the right reasoning structure can surface these capabilities even in smaller models.

## 3. Thought-MDP (Thought-by-Thought Generation)

We introduce thought-level MDPs, a framework that views LLM inference at the granularity of reasoning steps rather than tokens, providing structure for studying self-localization and self-correction. Let $p_\theta$ be an LLM with parameters $\theta$ and $x = (x[1], \ldots, x[n])$ a token sequence with $p_\theta(x) = \prod_{i=1}^{n} p_\theta(x[i] \mid x[1], \ldots, i-1)$. For reasoning tasks, $x$ denotes the context (prompt and question) and $y \sim p_\theta(\cdot \mid x)$ the response. We write $(u \oplus v)$ for concatenation. An MDP is characterized by $(\mathcal{S}, \mathcal{A}, \mathcal{P}, \mathcal{R})$: state space, action space, transition function, and reward. A token-level MDP defines states as $s_i = (x \oplus y[1] \oplus \ldots \oplus y[i-1])$, actions as single tokens $a_i = y[i]$, and deterministic transitions $s_{i+1} = (s_i \oplus a_i)$; a token-level policy is then $\pi_{\text{token}} : \mathcal{S} \to \mathcal{A}$. A thought-level MDP modifies only the action space: actions are now token sequences $a_j = (y_j[1], \ldots, y_j[k])$ for $k > 0$, yielding $\pi_{\text{thought}} : \mathcal{S} \to \mathcal{A}'$. We make no assumptions on the transition function beyond the Markovian property, taking $s_{j+1} = f(s_j, a_j)$; as a simple instanti-

ation, we use concatenation $s_{j+1} = (s_j \oplus a_j)$. The reward $\mathcal{R} : \mathcal{S} \rightarrow \{0, 1\}$ is defined only on terminal states $\mathcal{S}_{\text{term}} \subseteq \mathcal{S}$—states where the model produces an answer (e.g., \boxed{·}) or maximum depth is reached. In practice, we delimit thoughts using stop sequences the model emits at each step's end. Crucially, $\pi_{\text{token}}$ and $\pi_{\text{thought}}$ are induced by the same model $p_\theta$—only the prompting differs. Explicit thought boundaries may change how the model organizes reasoning into discrete, self-contained units; empirically, thought-level generation can yield higher initial accuracy before any correction (Figure 2), suggesting the structural difference may matter beyond enabling backtracking.

## 4. Thought-ICS: A Framework for Studying Self-Correction

We introduce **Thought-ICS** (Iterative Correction Sampling), a framework for studying iterative correction with LLMs with "structured generations"[1]. This framework comprises of two parts, a) *verification*, to determine if the generated response is correct and b) *correction*, which involves error localization and resampling from a partial generation (shared prefix). We illustrate Thought-ICS in Figure 1. Thought-ICS can be used to answer specific research questions by configuring each component to isolate specific capabilities. For example, we can use different sources of verification like: *self-verification*, when the model is asked to verify its generation, or *oracle-verification*, when the ground truth or a proxy for it is used a source of feedback. Similarly, localization granularity can be at the *token-level*, where the model is tasked to localize to a set of tokens or at the *thought-level* where the model localizes to a specific reasoning step; this can test whether LLMs can self-localize effectively and at what granularity. By varying different components, we can quantify different capabilities of the model. We describe main components of the Thought-ICS framework below.

**Thought-by-thought generation.** We use deliberate thought-level generation to generate an initial response, by taking actions in a thought-level MDP as formulated in Section 3. At each state $s$, we prompt the model to output the next action from the policy, $\pi_{\text{thought}}(a|s) = p_\theta(a \,|\, \text{prompt}(s))$ as defined above[2] and use the same system instruction[3] (i.e. prompt($\cdot$)) at every state. We provide

---

[1]Throughout the paper, we use the term structured generations to refer to reasoning traces which are generated thought-by-thought.

[2]For readers familiar with RL environments like OpenAI Gym (Brockman et al., 2016), this can be viewed as a step in a deterministic RL environment.

[3]System instruction: "*State your next reasoning step (one observation, calculation, or deduction),*" with each thought delimited by a closing marker.

---

**Algorithm 1** Thought-by-thought generation
---
1: **Input:** Question $x$, policy $\pi_{\text{thought}}$, max depth $D$
2: **Output:** Reasoning trace $\tau$
3: $s \leftarrow x$
4: **repeat**
5: $\quad a \sim \pi_{\text{thought}}(\cdot \mid s) \triangleright$ Sample next thought
6: $\quad s \leftarrow s \oplus a \triangleright$ Append thought to state
7: **until** $s$ contains answer or max depth reached
8: **return** $s$

---

formatting examples in-context, and at each step update the prompt to include the growing prefix of previously generated thoughts. See Appendix C.1 for details. The next state, $s'$ is taken to be the concatenation of the current state $s$ and the generated thought $a$. We set the termination condition to be the transition to a terminal state, i.e. when the model produces an answer (e.g. with a \boxed{·} marker) or the maximum depth is reached. More details are shown in Algorithm 1. In our implementation, we represent this process as a tree: each state is a node, and generating a thought adds a child node extending the current path. Initial generation traces a single root-to-leaf path, while the backtrack and resample steps (as explained below) branch the tree from an earlier node. A key benefit of this is that thoughts do not need to be retroactively parsed or segmented from a monolithic reasoning trace such as a chain-of-thought; with a Thought MDP, each complete thought/action is already a self-contained and segmented step.

**Verification.** Verification is a precursor to correction. In Thought-ICS, verification is used as a gating function: only responses which are identified to be incorrect continue iterative correction loop. The source of verification is configurable: in the fully autonomous setting, the model performs self-verification to judge its own generation; while in the oracle verification setting, an external source of feedback can be used to verify correctness of the model's generation.

**Localization.** The next step in Thought-ICS is error localization: to identify the error, given a response is identified as incorrect in the verification phase. In Thought-ICS, we only consider "self-localization" where the model (which generated the response) is prompted [4] to analyze its reasoning trace and identify the first erroneous thought in the sequence. For this, we aggregate the thought-level generation into one reasoning trace with each thought labelled separately. The model examines each thought for possible errors and returns both the thought number and reasoning if

---

[4]We use a simple prompt instructing the model to a) identify any logical flaws, arithmetic mistakes or incorrect assumptions in the generated reasoning trace, and b) return the incorrect step number along with reasoning if any erroneous step is found. See Appendix C.2 for the prompt we use.

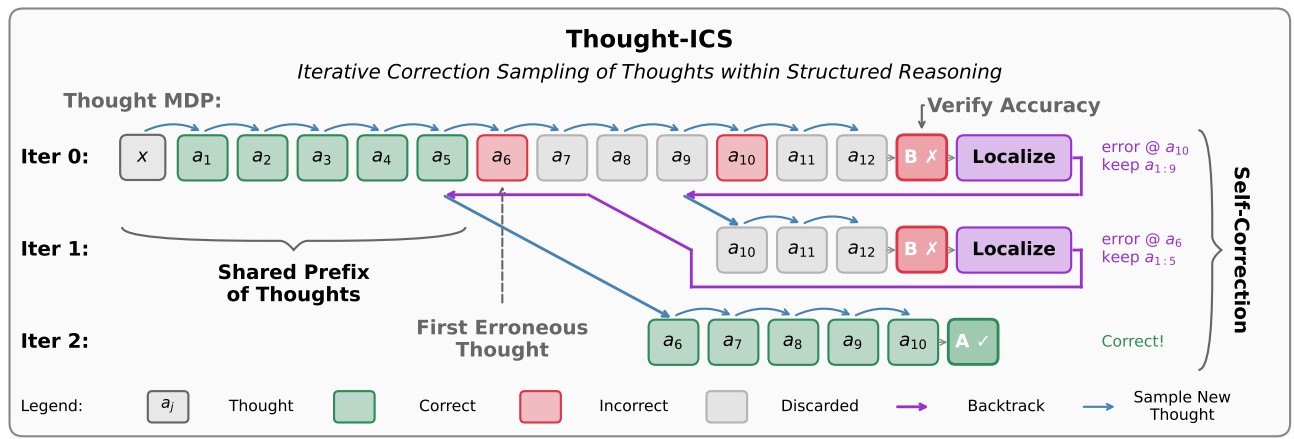

*Figure 1.* Illustration of the Thought-ICS framework. **Generation:** The model is tasked with sampling the next action/thought given a prefix of the original prompt and the previously sampled steps. We call this a Thought MDP, as it constructs a structured reasoning trace thought by thought $(a_1, a_2, \ldots)$ until it deems it has finished the task. **Verification:** A verification signal (configurable: self or external verification) indicates whether the final answer is correct or not; this gates whether the system continues or exits. **Localization:** The model is provided the full reasoning trace and asked to analyze each thought to identify the first thought where an error has been made. If it still cannot find an error, then it exits the loop. **Resampling:** Having identified an erroneous thought, the inference framework then backtracks to the last verified correct step, and resumes generation thought by thought from that shared prefix. This loop repeats until reasoning is verified as correct or a termination condition is met. In this example, the model corrects errors at steps 10 and 6 over two iterations before arriving at the correct answer. See Appendix A for the full correction example that this visualization corresponds to.

an erroneous thought is found.

**Resampling: Backtracking and regeneration.** In Thought-ICS, we implement a scaffolding to backtrack to the state corresponding to the erroneous thought and resample a completion using $\pi_{\text{thought}}(\cdot)$. To illustrate this, let $\tau = [s_e, a_e, a_{e+1}, \ldots]$ be a trajectory where $a_e$ is the incorrect thought as identified by the model in the error localization step. The resampling step in Thought-ICS backtracks to state $s_e$ and resamples a new completion conditioned on the *shared prefix* $\tau[: s_e]$. This is analogous to using resets in RL, where an intermediate state in a trajectory is taken to be the start state for a new one. Algorithm 2 illustrates one Thought-ICS loop: verification, localization, and resampling are performed iteratively for up to $L$ iterations until the verification step identifies the response to be correct or a termination condition is met. The framework has three such exit conditions: (1) VERIFIED ACCURACY, when the response is verified as correct (either by self or by external oracle); (2) V/L DISAGREEMENT, when verification flags an error but localization cannot find one; and (3) MAXITER, when maximum iterations are exhausted. We analyze these exit conditions in Section 6.

## 5. Self-Correction of Incorrect Reasoning

**Experiment Setup.** We experiment with eight instruction-tuned models spanning 3B–120B parameters across three open-source families: LLaMA 3 (3B, 8B, 70B) (Grattafiori et al., 2024), Qwen 2.5 (7B, 14B, 32B) (Yang et al.,

**Framework 2** Thought-ICS: Iterative Correction Sampling of Thoughts

1: **Input:** Question $x$, max iterations $L$
2: $\tau \leftarrow \text{Generate}(x)$      ▷ Thought-MDP (Alg. 1)
3: **repeat**
4:     **return** $\tau$ if Verify($\tau$)   ▷ (1) VERIFIED ACCURACY
5:     $e \leftarrow$ Localize first erroneous thought in $\tau$
6:     **return** $\tau$ if $e = \emptyset$    ▷ (2) V/L DISAGREEMENT
7:     $\tau_{\text{prefix}} \leftarrow \tau[: e]$     ▷ Backtrack to thought $e{-}1$
8:     $\tau \leftarrow \text{Generate}(\tau_{\text{prefix}})$     ▷ Resample correction
9: **until** max iterations
10: **return** $\tau$        ▷ (3) MAXITER

2024), and GPT-OSS (20B, 120B) (OpenAI et al., 2025). We evaluate on six reasoning benchmarks: AMC23 (kno, 2025) and AIME (Veeraboina, 2023) (math competition), MATH500-L5 (hardest tier of MATH) (Hendrycks et al., 2021), MathQA (arithmetic word problems) (Amini et al., 2019), CSQA (commonsense reasoning) (Talmor et al., 2019), and GPQA (graduate-level science) (Rein et al., 2023). These tasks are a mix of free-response and multiple choice questions; correctness is evaluated by exact match. We sample 100 problems per dataset, except AMC23 which contains 40 problems. Unless otherwise noted, figures show results averaged across all six datasets.

## 5.1. Structure Improves Self-Correction Under Oracle Verification

We begin by exploring whether language models can correct reasoning they know to be incorrect. Using an oracle verifier to filter for problems the model actually got wrong, we present the main result of this work: *models correct erroneous reasoning more often using structured reasoning traces sampled from a Thought MDP than using unstructured chain-of-thought traces.*

We instantiate two methods in the ICS framework, both using oracle verification. The first, Thought-ICS, generates structured reasoning by sampling thought-by-thought, where the model deliberates on the next semantically complete thought at each step. For localization, the model is provided the full sequence of thoughts and asked to output the incorrect thought number; structured generation already splits the reasoning trace into a chain of discrete steps that can be traversed during backtracking. The second, Token-ICS, generates reasoning traces using unstructured CoT. We use Token-ICS as a baseline to quantify the advantage of structured generation for self-correction. While one could attempt to impose structure on CoT using dedicated demarcation tokens (e.g., `<thought>`, `</thought>`), we find that small LLMs often exhibit inconsistent instruction following, making retroactive parsing of individual steps unreliable. As such, for localization in Token-ICS, the model is given the CoT trace and asked to quote the incorrect reasoning segment.

Recall that this work focuses on correction via localization, backtracking and resampling. Unlike prior works that regenerate solutions from scratch, where accuracy gains may simply reflect sampling variance, we want to test whether models can explicitly identify where they went wrong and resume from there. Hence, both Thought-ICS and Token-ICS use scaffolding to backtrack and resample from a "correct" prefix as identified in the localization step. As Fig. 2 shows, this scaffolding enables self-correction for both, though the gains are larger and more uniform for Thought-ICS, showing the advantage of structured generation for self-correction. Full results can be found in Tab. 4 in App. H.

## 5.2. Structure Enables Precise Self-Localization

We examine the localization step specifically: *can LLMs localize errors reliably in incorrect reasoning traces?* We find that LLMs self-localize more accurately with structured reasoning traces than with unstructured CoT traces, and that this gap grows with model scale. We define a successful localization as identifying the first erroneous thought in the sequence. When the model backtracks to before this thought, the resulting prefix is clean, free of any incorrect reasoning. An unsuccessful localization, by contrast, identifies a thought that was not the first error; the resulting prefix

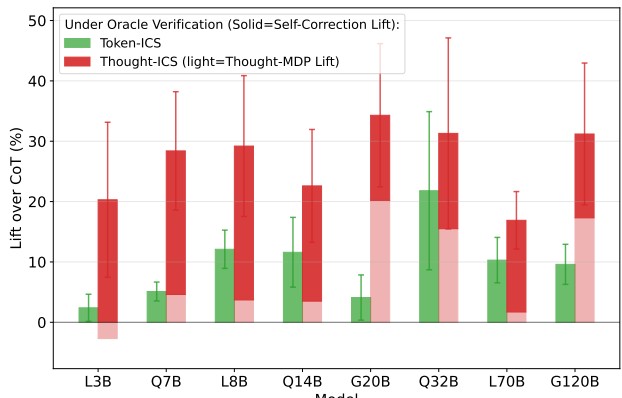

*Figure 2.* **Self-correction under an oracle verifier.** Localization on structured reasoning (Thought-ICS, red) achieves higher self-correction lift than within unstructured chain-of-thought (Token-ICS, green), *both operating on known incorrect solutions.* Light red shows initial lift from thought-by-thought generation (see App. D), solid red shows additional lift from self-correction (see Tab. 4, App. H).

remains erroneous, still containing the true first mistake.

**Establishing oracle localization.** To quantify self-localization abilities of LLMs, we construct an oracle localizer by using three frontier models (Sonnet-3.7, GPT-4.1 and GPT-5-mini) to independently identify the first erroneous step (for Thought-ICS) or the erroneous text (for Token-ICS) in each response. For Thought-ICS, when these three models agree on the error step, we take it as oracle error localization (Wang et al., 2024); in case of disagreement, we take the earliest error identified as a reasonable proxy. The three oracle models show high agreement: 51% unanimous, 74% within ±1 step, 85% within ±2 steps (see Fig. 14 in App. G). For Token-ICS, oracle agreement is lower since token counts are more granular than steps (a few hundred tokens on average), and because models may paraphrase or quote text with minor variations. Using a tolerance of 10% of the solution length, agreement reaches only 58%. Since this oracle is an LLM-as-judge consensus rather than ground truth, the deviation (self − oracle) and the clean versus erroneous prefix split we report below are defined relative to this reference; we use them to compare localization quality between structured and unstructured reasoning under a matched protocol.

**Models localize precisely within structured reasoning.** The discrete thought boundaries in structured generations enable precise localization. At each step, the model has made a deliberate decision about what the next complete thought should entail, creating natural decision boundaries where the structure clearly delineates where one thought ends and another begins. Fig. 3 shows that self-localization with Thought-ICS consistently obtains clean prefixes, with

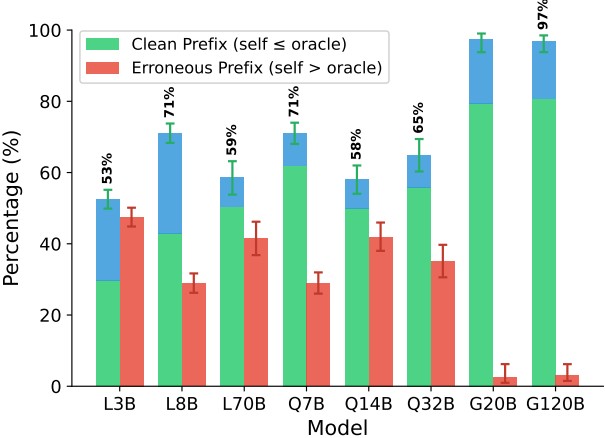

*Figure 3.* Self-localization within structure vs oracle localization. Clean prefix (left bar): self ≤ oracle, comprising exact matches (green) and earlier localizations (blue). Erroneous prefix (right bar, red): self > oracle.

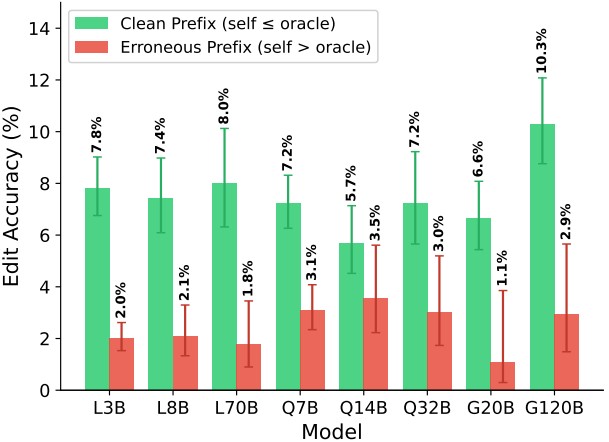

*Figure 4.* Self-correction accuracy within structure when resampling from clean prefixes (green) is 2–4× higher than from erroneous prefixes (red). Localization quality directly impacts correction performance within structure.

larger/capable models achieving more clean prefixes than erroneous ones. This analysis uses only cleanly-parsed localization outputs to isolate localization ability from format-following ability; smaller models often struggle with formatting instructions during localization, but we verify that findings remain consistent when including all localizations via fallback parsing as well (see App. G for details). The clean prefixes obtained from larger models are also more precise: a high percentage result in exact matches with the oracle localization as shown in the deviation distribution plot in Fig. 5. This plot also confirms that the correction accuracy lift for Thought-ICS is a result of precise localization and not conservatism (models localizing to the first step/first few steps). The deviation distribution for large models is concentrated around 0. Smaller models exhibit a wider deviation distribution with a right skew, implying that they miss the first error more often and obtain more erroneous prefixes

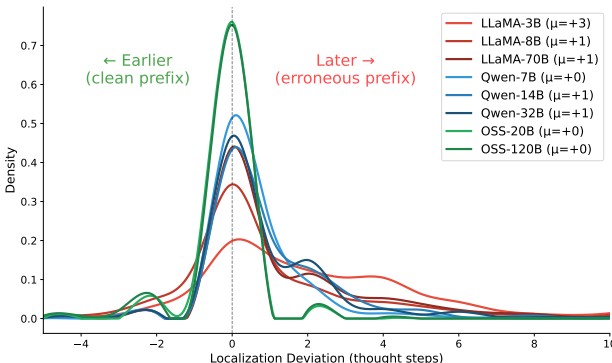

*Figure 5.* Distribution of localization deviation (self − oracle). Larger models show tighter distributions centred at zero, indicating more precise error localization with structured generation rather than LLMs being more conservative.

during localization, affecting downstream accuracy lift during correction. Recall that Thought-ICS is an inference time approach for self-correction; the models are not trained explicitly to be good at self-localization. It is worth noting that despite this, self-localization with thought-level generations meaningfully correlates with oracle localization.

**Precise localization yields more accurate resampling.** We find that resampling from a clean prefix improves the probability of correction (i.e. resampling to generate a correct response) by 2–4× over resampling from an erroneous prefix (Fig. 4). This analysis uses all localizations (including those requiring parsing fallbacks); we verify in App. G that restricting to cleanly-parsed localizations yields consistent findings as well. The mechanism is intuitive: an erroneous prefix which retains errors made by the model potentially derails it during resampling, harming self-correction. We illustrate the importance of resampling from a clean prefix in App. A. There, the model initially localizes to an erroneous step which isn't the first mistake in the reasoning. Resampling from this erroneous prefix fails and the model is unable to generate a correct response. In the next iteration, the model localizes to the first erroneous thought and succeeds in generating a correct response when resampling from this clean prefix.

**Without structure, localization is inconsistent.** As opposed to Thought-ICS, Token-ICS attempts the same localize-backtrack-and-resample approach but with unstructured chain-of-thought. Without discrete thought boundaries, models struggle to precisely attribute the error to a particular segment. We find that models tend to self-localize to segments later in the reasoning trace than what the oracle models identify, missing the root cause and resulting in

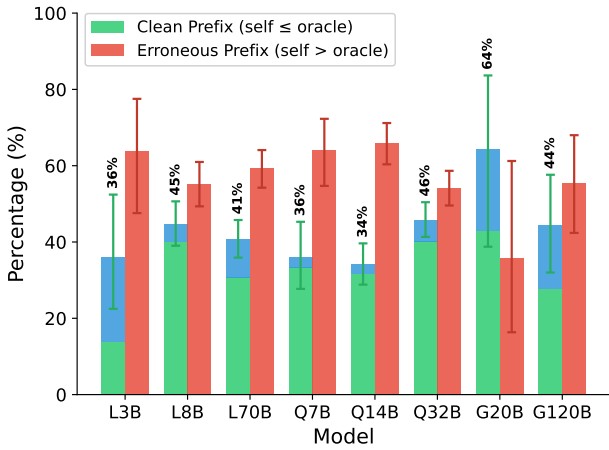

*Figure 6.* Self-localization within unstructured CoT. Without discrete thought boundaries, it is more difficult for the model to self-localize the exact erroneous step: only 30–45% yield clean prefixes, compared to 60–80% for Thought-ICS (Fig. 3).

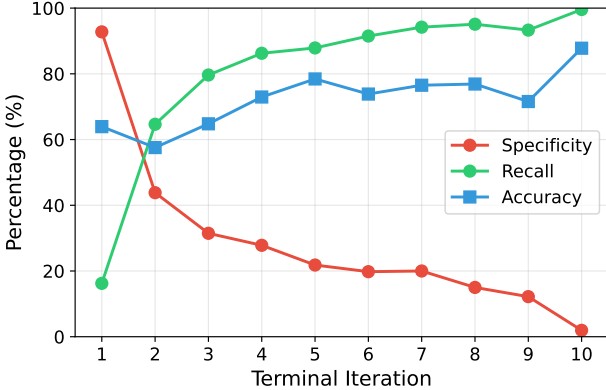

*Figure 7.* Self-verification metrics stratified by terminal iteration (the iteration at which the model accepts its response and exits). Specificity (number of correct responses left alone) collapses, while recall (number of incorrect responses flagged for correction) increases; models increasingly fail to recognize correct answers.

more erroneous prefixes than clean. This is shown in Fig. 6. As a consequence, Token-ICS achieves lower correction accuracy gains compared to Thought-ICS (see App. G for detailed analysis). The key takeaway: *structured generations make precise localization tractable, enabling more effective self-correction of incorrect reasoning.*

## 6. A Fully Autonomous Self-Correction System

Given the promising results in Sec. 5.1 showing that LLMs can self-localize with thought level generations, we ask: *can we develop an inference time method to do self-correction autonomously (without oracle verification)?* Self-verification is a key challenge towards developing such a system, and we propose some heuristics to overcome this.

### 6.1. Self-Verification Degrades Over Iterations

The first step in an autonomous self-corrective system is verification: determining whether the current response contains an accurate final answer. This step gates the correction loop: responses deemed correct exit the loop, while those flagged as incorrect go through the localization and backtrack-and-resample steps. We evaluate whether LLMs can reliably self-verify their own generations. Although LLMs appear to achieve reasonable self-verification accuracy at an aggregate level, having the knowledge to solve a problem does not confer the ability to verify the solution. We uncover interesting trade-offs which explain why aggregate accuracy is not helpful in downstream self-correction: essentially, the models are unable to accurately identify correct responses and end up breaking them after multiple iterations of self-correction.

We measure three metrics for self-verification: i) recall - fraction of incorrect responses flagged for self-correction,

ii) specificity - fraction of correct responses preserved, and iii) overall verification accuracy. We stratify the results by terminal iteration (the iteration at which the model accepts its response and exits), terminal iteration $n$ contains problems accepted after exactly $n$ correction attempts. Fig. 7 shows these metrics at different terminal iterations of the Thought-ICS algorithm, averaged across different models and datasets. Most models achieve good accuracy rates (between 60–80%) across iterations, with recall rates increasing almost monotonically. However, specificity collapses over iterations; the model becomes poor at identifying correct responses, i.e., correct responses remain in the loop and risk being broken. We find this recall-specificity trade-off to be a key reason for the poor self-verification abilities of LLMs.

**Self-correction under self-verification breaks more than it fixes.** We experiment with different self-verification methods in the Thought-ICS framework. In Tab. 1, we show that averaged across iterations, Thought-ICS with one self-verification call (labelled as **Single**) breaks more correct responses than it fixes incorrect responses, due to the trade-off in recall and specificity. We also test whether self-consistency approaches can help rectify this issue. For this, we sample 9 responses at each iteration and try a few different gating strategies: a) **Any**: which continues iterations for self-correction if any response is determined by the LLM to be incorrect, b) **Majority**: which does so if majority of the responses are determined to be incorrect, and c) **Unanimous**: which does so only if all the responses are determined to be incorrect. As shown in Tab. 1, all three approaches simply shift the recall-specificity trade-off without fixing it: the fraction of initially correct answers that were broken dominates the fraction of initially incorrect answers that were corrected after self-correction iterations.

| Method | Recall | Specificity | Broke | Fixed |
|---|---|---|---|---|
| Single | 68.3% | 66.9% | 10.9% | 6.2% |
| Any | 94.6% | 30.9% | 21.4% | 8.1% |
| Majority | 70.2% | 69.9% | 12.7% | 6.3% |
| Unanimous | 34.7% | 94.6% | 6.0% | 3.5% |

*Table 1.* Verification strategies of varying strictness (9 samples per iteration) trade recall for specificity, but all methods break more answers than they fix. Single: standard single-call verification. Any/Majority/Unanimous: flag for correction if any/majority/all samples indicate incorrect. Broke: % of initially correct answers broken. Fixed: % of initially incorrect answers corrected.

## 6.2. Confidence Safeguards Can Help Mitigate Self-Verification Failures

Recall from Sec. 4 that the Thought-ICS framework has 3 exit conditions. Based on our findings in Sec. 6.1, we dig deeper into specificity collapse over iterations and empirically identify two exit conditions with high break rates, as shown in Tab. 2. The high break rate (51%) with exit condition (2) V/L DISAGREEMENT reveals that in many cases, self-verification resulted in a false positive (model identifying a correct generation as incorrect), thereby breaking a correct response. Exit condition (3) MAXITER sees an even higher break rate (77%). We denote these two termination conditions as "low confidence corrections"; given the high break rates, the likelihood of meaningful self-correction is very low. When encountering these two termination conditions, instead of exiting with the last correction attempt as the answer, we reset to the initial response as the final answer. When the model normally self-verifies its reasoning as correct in exit condition (1) VERIFIED ACCURACY, it is able to achieve net positive lift, so there we keep the final corrected response. We denote this method as Thought-ICS-A. Fig. 8 shows how this confidence safeguard mitigates breakage, resulting in an inference-time method that can fix more than it breaks per correction iteration.

| Termination Condition | Broke | Fixed | Net Lift |
|---|---|---|---|
| (1) VERIFIED ACCURACY | 2.5% | 6.4% | +3.9% |
| (2) V/L DISAGREEMENT | 51.3% | 2.0% | −49.3% |
| (3) MAXITER | 77.4% | 5.4% | −72.0% |

*Table 2.* Self-correction performance by termination condition. Thought-ICS has 3 exit criteria: (1) VERIFIED ACCURACY, (2) V/L DISAGREEMENT, or (3) MAXITER. (1) achieves positive lift, while (2) and (3) show high break rates.

## 6.3. Thought-ICS-A as an Autonomous Inference-Time Correction Method

We compare the overall performance of Thought-ICS-A as an inference time method for autonomous self-correction

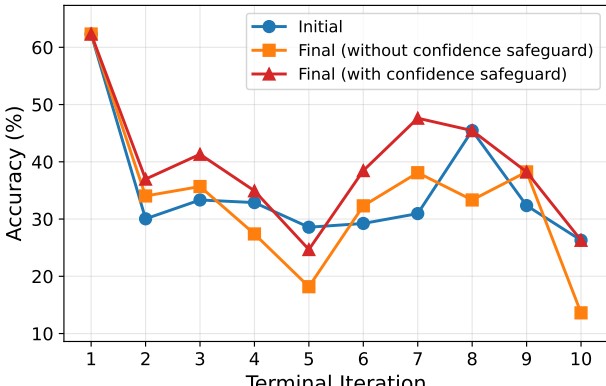

*Figure 8.* Iteration-by-iteration correction lift of Thought-ICS-A over initial Thought-MDP accuracy (blue). Without confidence safeguard in Sec. 6 in the verification, self-correction largely breaks more than it fixes over iterations (orange). With confidence safeguard, the system mitigates many cases of breakage (red), achieving tangible net positive correction lift at nearly every iteration.

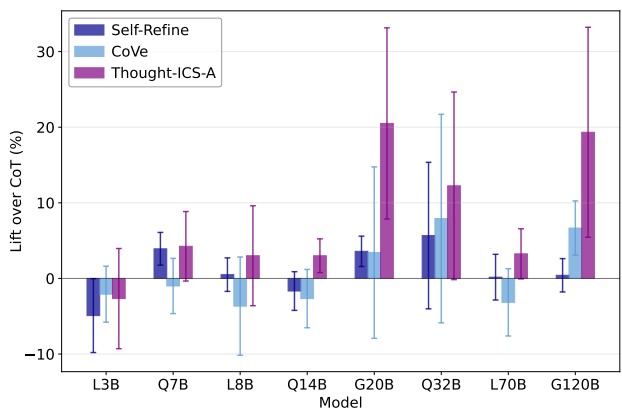

*Figure 9.* Self-correction lift in fully autonomous setting, pooled across all six datasets. Thought-ICS-A outperforms Self-Refine and CoVe across models. Full per-dataset results in Tab. 7.

against comparable baselines which use CoT generations: **Self-Refine** (Madaan et al., 2023), which generates feedback on errors and resamples from scratch; and **CoVe** (Dhuliawala et al., 2023), which generates verification questions to check specific steps before regenerating. See App. E for full implementation details of each of these methods. Fig. 9 shows self-correction lift averaged over all six datasets. As can be seen, Thought-ICS-A outperforms both baselines across all models, except Llama-3B.

## 7. Conclusion and Limitations

We introduced Thought-ICS, a framework for studying self-correction in LMs through structured reasoning. Our key finding is that structure improves self-correction over unstructured CoT: generating reasoning thought-by-thought in a Thought MDP enables models to self-localize errors at step

granularity and correct them via backtracking and resampling. With oracle verification, this yields 20–40% accuracy gains across models and benchmarks. For fully autonomous correction, self-verification remains the bottleneck, but we show how effective system design can achieve positive net correction lift. Our evaluation focuses on models in the 3-120B parameter range and on math, science, and commonsense reasoning benchmarks with verifiable answers, leaving larger frontier models and open-ended domains for future work. It remains unclear whether such methods constitute genuine self-correction or succeed through structured resampling. Our localization oracle is a frontier-model consensus available at the time of writing, not a ground-truth annotation, so the resulting metrics (the $\text{self} - \text{oracle}$ deviation and the clean versus erroneous prefix split) are defined with respect to this reference and support a relative comparison between structured and unstructured reasoning rather than absolute accuracy. The localization step itself can also be improved, through further prompt engineering, confidence-based methods, or explicit training for self-localization. As for designing a fully autonomous self-correction system, the bottleneck remains self-verification, motivating future work on deliberately training models to self-verify. Nonetheless, our results on verified incorrect reasoning reveal large potential gains as verification capabilities improve.

## Impact Statement

This paper presents work on improving language model reasoning through self-correction at the thought level. While our primary goal is to advance the field of Machine Learning, we note that enabling models to identify and correct their own reasoning errors could improve performance on complex reasoning tasks and reduce reliance on external verification. As with any capability improvement, appropriate safeguards remain important during deployment.

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

## A. Thought-ICS: Tree Visualization Example

Fig. 10 shows the full reasoning correction trace for the example in Fig. 1. The problem (GPQA #71) is given as follows.

**Problem.** Given that particles with Lorentz factor $\gamma = 20$ have a 1/3 survival rate reaching a detector, what Lorentz factor is needed for 2/3 to survive?

**Iteration 0: Initial attempt (error-localization in step 10).** The model begins with correct setup in steps 1–5: identifying that 2/3 of particles decay before reaching the detector, noting the detector radius (30m), and establishing notation. However, in steps 6–9, the model makes a conceptual error: it sets up the inequality backwards, and reasons about when decay time is shorter than travel time rather than using exponential decay properly. This flawed reasoning propagates to step 10, where the model incorrectly concludes $\gamma \approx 40$ and selects answer (B). Upon self-verification, the model detects the final answer is wrong, and localizes the error to step 10.

**Iteration 1: Backtrack to step 9, regenerate (error-localization in step 6).** After backtracking to step 9, the model regenerates steps 10–14 but arrives at the same wrong answer (B). This time, self-verification traces the error back further to step 6, recognizing that the fundamental approach was flawed: the inequality setup itself was incorrect, not just the final calculation.

**Iteration 2: Backtrack to step 5, regenerate (succeeds).** With a deeper backtrack to step 5, the model takes a completely different approach in steps 6–9. It correctly applies the exponential decay law: $P(\text{survive}) = \exp(-t_{\text{proper}}/\tau)$ where $t_{\text{proper}} = t/\gamma$. Using the given condition that $\gamma = 20$ yields $P = 1/3$, it derives $r/(c\tau) = 20 \cdot \ln(3)$. For $P = 2/3$, solving $\exp(-r/(\gamma c\tau)) = 2/3$ gives $\gamma = 20 \cdot \ln(3)/\ln(3/2) \approx 54$. The model correctly selects answer A and self-verification confirms the reasoning is sound.

This example illustrates how Thought-ICS can recover from errors at different depths: a shallow error (step 10) is caught first, but when regeneration reveals the error was actually deeper (step 6), the method backtracks further until finding a correct reasoning path.

**Importance of accurate localization.** If localization identifies a step after the true error, the erroneous reasoning remains in the shared prefix and can derail the resampled continuation. In this example, Iteration 1 fails because step 6 (the true error) was still present in the prefix when regenerating from step 9. Only when Iteration 2 backtracks to step 5, before the flawed inequality setup, does the model find a correct path. Fig. 4 quantifies this effect across all experiments.

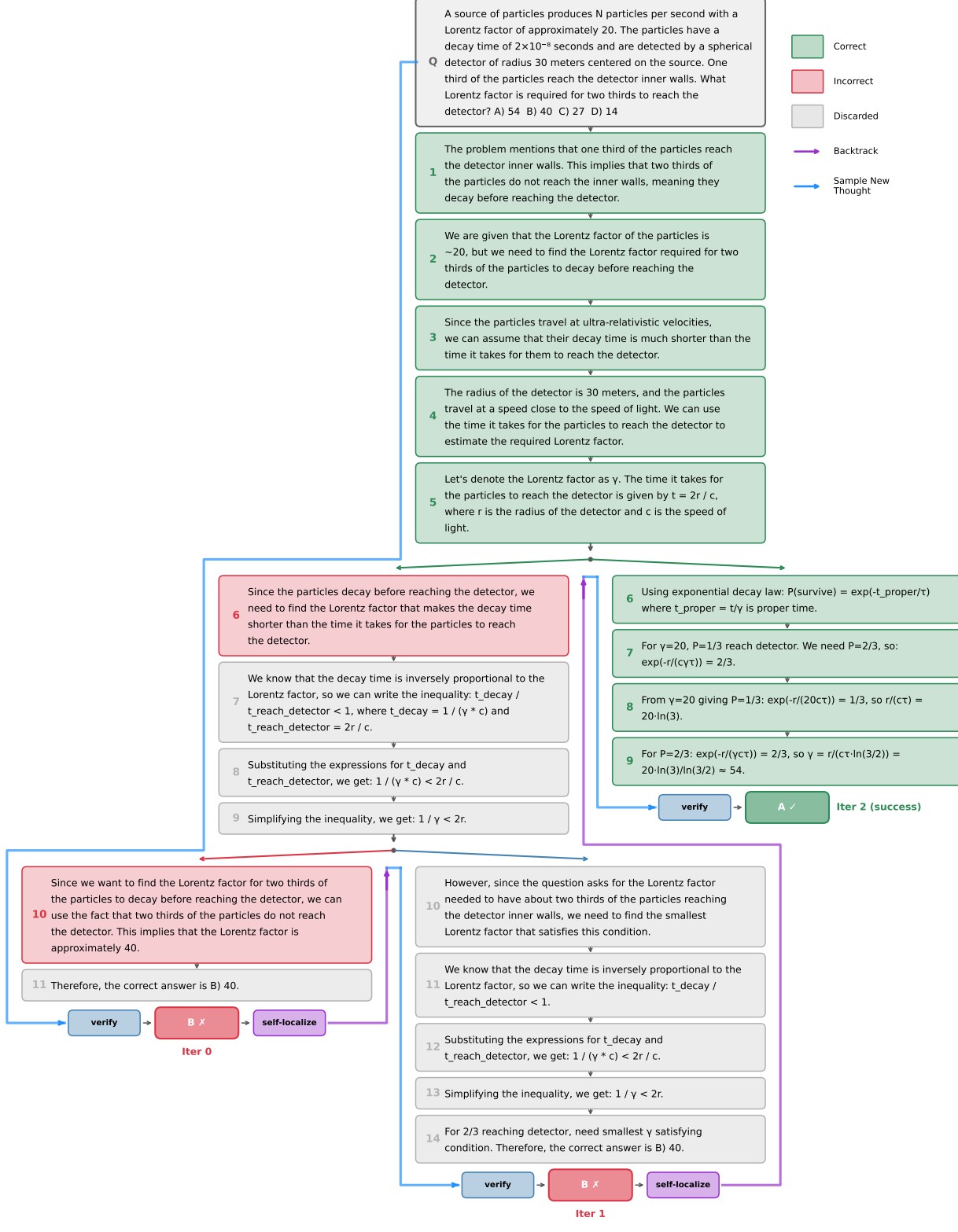

*Figure 10.* Full reasoning correction trace for the GPQA particle decay problem. The tree shows three iterations: Iteration 0 fails at step 10 (shallow error), Iteration 1 fails at step 6 (deeper error discovered), and Iteration 2 succeeds with correct exponential decay reasoning. Green = kept steps, red = error steps, gray = discarded steps.

# B. CoT vs Thought-ICS: Comparative Examples

We present concrete examples comparing standard Chain-of-Thought (CoT) with Thought-ICS on the same problems. These examples illustrate two scenarios: (1) when both methods arrive at the correct answer, and (2) when Thought-ICS successfully corrects an error that CoT alone cannot.

## B.1. Example 1: Both Methods Succeed (AMC23 #1)

**Problem.** Cities $A$ and $B$ are 45 miles apart. Alicia lives in $A$ and Beth lives in $B$. Alicia bikes towards $B$ at 18 miles per hour. Leaving at the same time, Beth bikes toward $A$ at 12 miles per hour. How many miles from City $A$ will they be when they meet?

**CoT Response.**

> To find out how many miles from City $A$ they will meet, we need to determine the time it takes for them to meet and then calculate the distance Alicia travels in that time. Let $t$ be the time in hours. In $t$ hours, Alicia travels $18t$ miles and Beth travels $12t$ miles. Since they bike towards each other: $18t + 12t = 45$, so $30t = 45$, giving $t = 1.5$ hours. Distance from $A = 18 \times 1.5 = \boxed{27}$ miles.

**Thought-ICS Response (Iteration 0).**

> **Step 1:** To find out how many miles from City $A$ they will meet, we need to determine the time it takes for them to meet and then calculate the distance Alicia travels in that time.</thought>
>
> **Step 2:** Let's denote the time it takes for Alicia and Beth to meet as $t$ hours. In $t$ hours, Alicia will have traveled $18t$ miles and Beth will have traveled $12t$ miles. Since they are biking towards each other, the total distance they cover together is 45 miles. Therefore: $18t + 12t = 45$.</thought>
>
> **Step 3:** Simplifying: $30t = 45$. Solving for $t$: $t = 1.5$ hours.</thought>
>
> **Step 4:** Now, we need to find how far Alicia has traveled in 1.5 hours. Distance $= 18 \times 1.5 = 27$ miles.</thought>
>
> **Step 5:** Therefore, they will meet 27 miles from City $A$. The answer is $\boxed{27}$.</thought>

**Analysis.** Both methods produce correct reasoning and arrive at the answer 27. Thought-ICS self-verification confirms the solution is correct, so no correction is needed. This demonstrates that the structured format does not impede performance when the initial reasoning is sound.

## B.2. Example 2: Thought-ICS Corrects Error (AMC23 #5)

**Problem.** In a table tennis tournament every participant played every other participant exactly once. Although there were twice as many right-handed players as left-handed players, the number of games won by left-handed players was 40% more than the number of games won by right-handed players. (There were no ties and no ambidextrous players.) What is the total number of games played?

**CoT Response.**

> Let $r$ be the number of right-handed players and $l$ be the number of left-handed players. We have $r = 2l$. The total number of games is $\binom{r+l}{2}$. Let $R$ be games won by right-handed players and $L$ be games won by left-handed. We have $L = 1.4R$. Each game has one winner, so $R + L = \binom{r+l}{2}$. Substituting $L = 1.4R$: $2.4R = \binom{3l}{2}$. Testing $l = 8$: games $= \binom{24}{2} = 276$. We check: $R = 115$, $L = 161$. This works, so the answer is $\boxed{276}$.

**Ground truth:** 36. **CoT is incorrect.**

**Thought-ICS Response.** *Iteration 0* (fails):

> **Steps 1–9:** [Similar reasoning to CoT, arrives at $l = 8$. Final answer is 276]
>
> **Self-localization:** Identifies error at Step 1: the equation setup is flawed because it conflates games won with games played.

*Iteration 1* (succeeds, regenerated from empty prefix):

> **Step 1:** Let the number of left-handed players be $l$ and the number of right-handed players be $2l$. The total number of players is $3l$.</thought>
>
> **Step 2:** The total number of games played is $\binom{3l}{2} = \frac{3l(3l-1)}{2}$.</thought>
>
> **Step 3:** Let $r$ be games won by right-handed players and $l_w$ be games won by left-handed players. We have $l_w = 1.4r$.</thought>
>
> **Step 4:** Total games won $= r + l_w = 2.4r = \frac{3l(3l-1)}{2}$.</thought>
>
> **Step 5:** Solving for $r$: $r = \frac{5l(3l-1)}{8}$. For $r$ to be an integer, $5l(3l-1)$ must be divisible by 8.</thought>
>
> **Step 6:** Testing $l = 3$: $5 \cdot 3 \cdot 8 = 120$, divisible by 8. So $l = 3$, $2l = 6$, total players = 9.</thought>
>
> **Step 7:** Total games $= \binom{9}{2} = 36$.</thought>
>
> **Step 8:** Therefore, the total number of games played is $\boxed{36}$.</thought>

**Analysis.** The initial attempt made an error in the algebraic setup (Step 1), leading to an incorrect answer. Self-verification detected the error, self-localization identified Step 1 as the source, and regeneration from the empty prefix produced a corrected solution with tighter integer constraints, arriving at the correct answer 36.

### B.3. Example 3: Thought-ICS Corrects Arithmetic Error (MATH)

**Problem.** Find the product $CD$ of the integers $C$ and $D$ for which $\frac{C}{x-3} + \frac{D}{x+8} = \frac{4x-23}{x^2+5x-24}$ for all real values of $x$ except $-8$ and 3.

**CoT Response.**

> The equation must hold for all $x$ except $-8$ and 3. The numerators must be equal: $4x-23 = C(x+8)+D(x-3)$. When $x = 3$:
>
> $11C = -11$, so $C = -1$. When $x = -8$: $-11D = -59$, so $D = 59/11$. Therefore $CD = -1 \times (59/11) = \boxed{-\frac{59}{11}}$.

**Ground truth:** $-5$. **CoT is incorrect.**

**Thought-ICS Response.** *Iteration 0* produces the same error (non-integer $D$).

*Self-verification* identifies the error at Step 4: "The equation is not being used correctly. The correct approach is to equate coefficients."

*Iteration 1* (regenerated from Step 3):

> **Step 4:** Equating coefficients of $x$: $4 = C + D$.</thought>
>
> **Step 5:** Equating constant terms: $-23 = 8C - 3D$.</thought>
>
> **Step 6:** From $C + D = 4$: $D = 4 - C$. Substituting: $-23 = 8C - 3(4 - C) = 11C - 12$.</thought>
>
> **Step 7:** Solving: $-11 = 11C$, so $C = -1$. Then $D = 4 - (-1) = 5$.</thought>
>
> **Step 8:** Therefore $CD = (-1)(5) = \boxed{-5}$.</thought>

**Analysis.** The initial approach (substituting specific $x$ values) led to a non-integer solution, violating the problem constraint that $C$ and $D$ are integers. Self-verification caught this inconsistency, and the corrected solution used coefficient matching to find integer solutions.

## C. Thought-ICS: Method details

We give implementation details for Thought-ICS below, including detailed prompts.

### C.1. Thought-level generation

The prompt we use for thought-by-thought generation (Algorithm 1) is shown below.

```
"""
You are solving a problem step-by-step.

Instructions:

1. State your next reasoning step (one observation, calculation, or
   deduction)

2. End each thought with </thought>

3. Continue until you reach the final answer, then write it in
   \boxed{answer} format

Examples:

Q: In how many ways can 5 distinct books be arranged on a shelf if 2
specific books must not be adjacent?

1. Total arrangements without restrictions is 5!  = 120</thought>

2. I need to subtract arrangements where the 2 specific books ARE
   adjacent</thought>

3. If I treat the 2 books as a single unit, I have 4 units to arrange:  4!
   = 24 ways</thought>

4. The 2 books within their unit can be arranged in 2!  = 2 ways</thought>

5. So arrangements with the books adjacent = 24 × 2 = 48</thought>

6. Therefore, arrangements where they are NOT adjacent = 120 − 48 =
   \boxed{72}</thought>

Q: A rectangle has area 48 and perimeter 28.  What is the length of its
diagonal?

1. Let length = l and width = w.  From the area:  lw = 48</thought>

2. From the perimeter:  2l + 2w = 28, so l + w = 14</thought>

3. From l + w = 14, we get w = 14 − l.  Substituting into lw = 48:  l(14 −
   l) = 48</thought>

4. Expanding:  14l − l² = 48, so l² − 14l + 48 = 0.  Factoring:  (l − 6)(l −
   8) = 0</thought>

5. So l = 8 and w = 6 (or vice versa).  Using the Pythagorean theorem:  d²
   = 8² + 6² = 64 + 36 = 100</thought>

6. Therefore d = 10, so the answer is \boxed{10} </thought>

Q: {<text of the question>}

Append thoughts in the current state, if any

1. Thought 1

2. Thought 2
        ⋮
"""
```

## C.2. Self-localization

**Formatting thoughts for verification.** Before prompting for self-localization, we convert the structured thought sequence back into a numbered list format. Given thoughts $(a_1, a_2, \ldots, a_n)$, we format them as:

```
Step 1: <content of a_1>
Step 2: <content of a_2>
...
Step n: <content of a_n>
```

This explicit numbering allows the model to reference errors by step number in its response. The model is then prompted to identify the first erroneous step (if any) and return both the step number and its reasoning.

```
"""
You are given a reasoning trace: {<insert reasoning chain here>}

Carefully verify your reasoning chain step by step. If you identify any
errors (logical flaw, arithmetic error, or incorrect assumption), determine
which step number (1 to {<length of reasoning trace>}) contains the first
critical error.

Also provide your reasoning. Then conclude with:

 1. \boxed{step_number} if you found an error

 2. \boxed{0} if the reasoning is correct

"""
```

## C.3. Resampling from prefix

When an error is localized to step $j$, we truncate the reasoning chain to the prefix $(a_1, \ldots, a_{j-1})$ and regenerate from that point. The resampling step reuses the same prompt template from Step 1 (Section C.1), with the validated prefix appended to the prompt. This ensures the model continues in the same structured format while exploring an alternative reasoning path.

## C.4. Thought extraction and parsing

**Models naturally comply with the thought-by-thought format.** Instruction-tuned models reliably follow the thought-by-thought format without additional training. Given the prompt template with few-shot examples (Section C.1), models consistently emit the `</thought>` delimiter at the end of each reasoning step, produce semantically coherent thoughts, and naturally terminate by producing a \boxed{} answer. This compliance emerges purely from in-context learning across all model families (LLaMA, Qwen, GPT-OSS) at all scales (3B–120B). Prior work has documented that structured output requirements can degrade reasoning performance (Tam et al., 2024; Lou et al., 2024), but our format avoids this by asking the model to generate one thought at a time rather than formatting an entire multi-step response upfront. This is also preferable to retroactively parsing unstructured CoT into discrete steps, where sentence boundaries do not reliably correspond to logical reasoning steps and the model has no awareness of step boundaries during generation. With the Thought MDP, the model explicitly decides when each thought is complete by emitting the delimiter, enabling localization to reference step numbers directly and resampling to preserve complete reasoning units.

**Implementation details.** For thought-level generation, we use `</thought>` and double newline as stop sequences; for unstructured CoT (Token-ICS, Self-Refine, CoVe, Iterative-CoT), we use only the model's natural stopping behavior. We use progressive token budgets (150, 300, 500 tokens) to handle varying thought lengths without truncation. A state is terminal when it contains a \boxed{} answer or reaches maximum depth $D = 100$. Edge cases: invalid step numbers $j > n$ are treated as $j = n$; missing delimiters result in treating the entire output as a single thought; step numbers are extracted from \boxed{·} with fallback to the last integer in the response.

## C.5. Additional details

**Inference.** All experiments use vLLM (Kwon et al., 2023) for inference with tensor parallelism across available GPUs. Sampling parameters: temperature = 0.5, top-$p$ = 0.9, top-$k$ = 50. Maximum tokens: 2048 for solution generation, 512 for verification questions and answers, 1024 for feedback generation, 150 per thought for structured generation.

# D. Initial Accuracy: Token MDP vs Thought MDP

A natural concern is whether imposing thought-level structure degrades initial reasoning quality compared to standard token-level generation; prior work has shown that complex formatting requirements can hurt reasoning performance, especially in smaller models (Tam et al., 2024). Table 3 compares iteration 0 accuracy (before any self-correction) across both MDP formulations.

We find that the Thought MDP does not degrade initial accuracy and often improves it: +6.5% on average, with gains in 36 of 48 model-dataset pairs. This demonstrates that our incremental thought-by-thought format provides the structural benefits needed for self-correction (Section 3) without imposing a reasoning tax. For larger models (20B+), the gains are substantial on difficult reasoning tasks (AIME: +35–41%, AMC23: +18–30%), suggesting that explicit thought boundaries may help capable models organize their deliberation more effectively.

*Table 3.* Initial accuracy (%): Token MDP vs Thought MDP generation. Both measure iteration 0 accuracy before self-correction. $\Delta$ = Thought $-$ Token. Structured thought-by-thought generation improves initial accuracy by +6.5% on average.

| Dataset | | 3B | 7B | 8B | 14B | 20B | 32B | 70B | 120B | Avg |
|---------|---------|-----|-----|-----|-----|-----|-----|-----|------|------|
| AMC23 | Token | 15 | 38 | 8 | 45 | 50 | 40 | 60 | 58 | 39 |
| | Thought | 18 | 42 | 20 | 48 | 80 | 68 | 62 | 75 | 52 |
| | $\Delta$ | +2 | +5 | +12 | +2 | +30 | +28 | +2 | +18 | +12.5 |
| AIME | Token | 7 | 16 | 4 | 18 | 21 | 18 | 39 | 24 | 18 |
| | Thought | 5 | 15 | 4 | 20 | 61 | 53 | 41 | 65 | 33 |
| | $\Delta$ | -2 | -1 | 0 | +2 | +40 | +35 | +2 | +41 | +14.6 |
| MATH-L5 | Token | 15 | 35 | 8 | 40 | 45 | 41 | 39 | 42 | 33 |
| | Thought | 9 | 39 | 15 | 42 | 57 | 54 | 37 | 54 | 38 |
| | $\Delta$ | -6 | +4 | +7 | +2 | +12 | +13 | -2 | +12 | +5.2 |
| CSQA | Token | 69 | 71 | 70 | 84 | 77 | 89 | 85 | 84 | 79 |
| | Thought | 59 | 85 | 67 | 90 | 78 | 90 | 88 | 79 | 80 |
| | $\Delta$ | -10 | +14 | -3 | +6 | +1 | +1 | +3 | -5 | +0.9 |
| GPQA | Token | 20 | 26 | 21 | 35 | 27 | 42 | 53 | 45 | 34 |
| | Thought | 22 | 27 | 23 | 40 | 45 | 43 | 56 | 66 | 40 |
| | $\Delta$ | +2 | +1 | +2 | +5 | +18 | +1 | +3 | +21 | +6.6 |
| MathQA | Token | 48 | 69 | 57 | 84 | 71 | 86 | 73 | 79 | 71 |
| | Thought | 36 | 71 | 44 | 82 | 82 | 89 | 67 | 91 | 70 |
| | $\Delta$ | -12 | +2 | -13 | -2 | +11 | +3 | -6 | +12 | -0.6 |

Model sizes: 3B = LLaMA-3.2-3B, 7B = Qwen2.5-7B, 8B = LLaMA-3.1-8B, 14B = Qwen2.5-14B, 20B = GPT-OSS-20B, 32B = Qwen2.5-32B, 70B = LLaMA-3.1-70B, 120B = GPT-OSS-120B.

# E. Baseline Methods

## E.1. Self-Refine

Self-Refine (Madaan et al., 2023) operates through a generate-feedback-refine loop. We follow the evaluation protocol of Huang et al. (2024), using generic feedback prompts without task-specific information (which can artificially inflate performance). Self-Refine runs for up to 10 iterations, regenerating from scratch each time, unlike Thought-ICS which preserves a validated prefix.

**Generation.** Uses a standard CoT prompt:

```
Solve the following problem step by step.  Show your reasoning clearly and
put your final answer in \boxed{answer}.
Problem:  {<problem text>}
Solution:
```

**Feedback.** The model reviews its solution and provides error analysis:

```
You are reviewing a solution to a math problem.  Analyze it carefully for
errors.
Problem:  {<problem text>}
Solution to review:  {<solution text>}
Provide feedback on this solution:

 1. Is the solution correct?  Answer YES or NO at the start.

 2. If NO, identify the specific error(s) and explain what went wrong.

 3. If YES, confirm the solution is correct.

Feedback:
```

**Refinement.** If errors are identified, the model regenerates from scratch using the feedback:

```
You previously attempted to solve a problem but received feedback indicating
errors.  Use the feedback to produce a corrected solution.
Problem:  {<problem text>}
Your previous solution:  {<previous solution>}
Feedback on your solution:  {<feedback text>}
Now provide a corrected solution.  Show your reasoning step by step and put
your final answer in \boxed{answer}.
Corrected Solution:
```

### E.2. Chain-of-Verification (CoVe) details

CoVe (Dhuliawala et al., 2023) was originally designed to reduce hallucinations in factual question-answering tasks (e.g., list-based queries, long-form generation). We adapt it to mathematical reasoning because it performs explicit step-level verification: the model generates verification questions targeting specific steps in its solution, making it a natural comparison point for our thought-level approach.

**Why factored verification.** The original CoVe paper proposes three variants: Joint (verification questions answered with baseline context), 2-Step (questions planned jointly, answered independently), and Factored (both planning and answering done independently). We implement the Factored approach because it prevents the model from simply repeating its original reasoning when answering verification questions. By answering each question in isolation (without seeing the baseline solution), the model is more likely to catch errors rather than rationalize them. This makes CoVe a stronger baseline for comparison.

**Single iteration by design.** Unlike Self-Refine and Thought-ICS which iterate until convergence or a maximum budget, CoVe performs exactly one refinement pass. The model generates a baseline solution, plans 2–4 verification questions, answers each independently, and produces a single final response incorporating the verified information. We preserve this design choice from the original paper rather than extending CoVe to multiple iterations, as the verification question mechanism is designed for a single correction pass.

**Baseline generation.** Uses a standard CoT prompt:

```
Solve the following problem step by step.  Show your reasoning clearly and
put your final answer in \boxed{answer}.
Problem:  {<problem text>}
Solution:
```

**Plan verification questions.**   The model generates questions to verify its solution:

```
You have provided a solution to a math problem.  Now generate verification
questions to check if your solution is correct.
Problem:  {<problem text>}
Your solution:  {<baseline solution>}
Generate 2-4 specific verification questions that would help verify the
correctness of this solution.  Each question should check a specific step,
calculation, or reasoning in the solution.
Format your questions as a numbered list:

1. [First verification question]

2. [Second verification question]

3. ...

Verification Questions:
```

**Execute verification.**   Each question is answered independently (without the baseline response context):

```
Answer the following question carefully and precisely.
Question:  {<verification question>}
Answer:
```

**Final verified response.**   The model produces a corrected solution using the verification results:

```
You previously solved a problem and generated verification questions with
answers.  Use this verified information to produce a final, corrected
solution.
Problem:  {<problem text>}
Your baseline solution:  {<baseline solution>}
Verification questions and answers:  {<Q1:  ...  A1:  ...  Q2:  ...  A2:
...>}
Based on the verification results, provide your final solution.  If the
verifications revealed any errors, correct them.  Show your reasoning step
by step and put your final answer in \boxed{answer}.
Final Solution:
```

### E.3. Token-ICS

Token-ICS applies the same iterative correction framework as Thought-ICS but operates on unstructured CoT traces. Instead of identifying an erroneous step number, the model quotes the exact text where the error occurs. When quote matching succeeds, we truncate at that position and regenerate; when it fails, we regenerate from scratch. Token-ICS runs for up to 10 iterations.

**Generation.**   Uses a standard CoT prompt:

```
Solve the following math problem step by step.  Show your reasoning clearly,
then provide your final answer in the format \boxed{answer}.
Problem:  {<problem text>}
Solution:
```

**Self-localization.**    Rather than outputting a step number, Token-ICS prompts the model to quote the erroneous text verbatim:

```
Problem:  {<problem text>}
Current solution:  {<solution text>}
Carefully verify your solution step by step.  If you identify any errors
(logical flaw, arithmetic error, or incorrect assumption), quote the EXACT
text (word-for-word) where the first critical error occurs.  This should be
a continuous excerpt from your solution above.
Provide your reasoning and analysis.  Then conclude with:

  • \boxed{ERROR_QUOTE: "exact text from solution where error occurs"} if you
    found an error
  • \boxed{CORRECT} if the solution is correct
```

**Resampling.**    When an error quote is extracted, we use string matching to find the quoted text in the solution and truncate at that position. The model then continues from the truncated prefix using the same generation prompt. If the quoted text cannot be found (due to minor misquotes), we regenerate the full solution from scratch.

### E.4. Answer extraction and evaluation

**Answer extraction.**    We extract final answers from the `\boxed{answer}` format, taking the last occurrence when multiple boxed expressions appear (e.g., when the model revises its answer). For nested braces, we use brace counting to find the matching close brace.

**Answer matching.**    For multiple-choice datasets (CSQA, GPQA, MathQA), we normalize answers (lowercase, strip whitespace) before comparison. For math datasets (MATH500, AIME, AMC23), we use exact string matching.

## F. Self-Verification Analysis

This section provides additional figures on self-verification performance. See Sec. 6.1 for discussion of the recall-specificity tradeoff and specificity collapse.

Figure 11 shows self-verification metrics broken down by model; models within the same family behave similarly, and scale does not consistently improve verification. Figure 12 shows confusion matrix components across iterations: true negatives decrease while false positives rise, reflecting specificity collapse. Figure 13 shows the distribution of problems by terminal iteration; most problems exit early, and those surviving to later iterations are systematically harder.

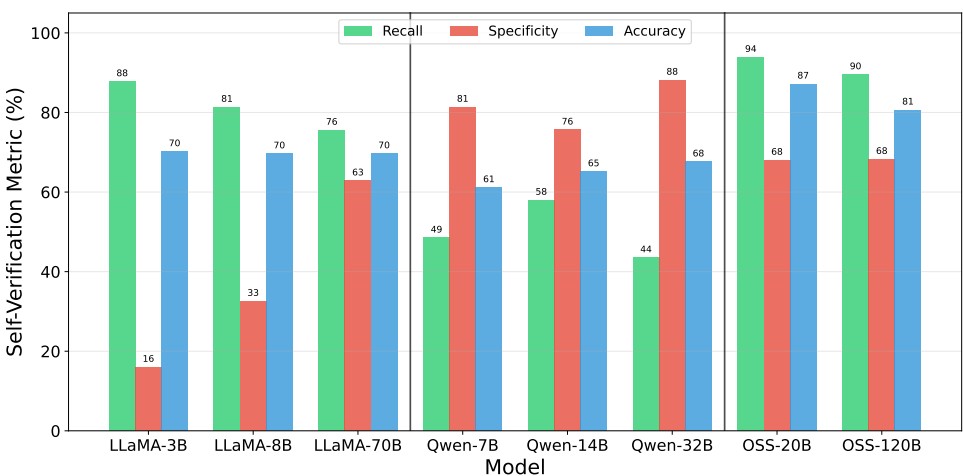

*Figure 11.* Self-verification metrics by model.

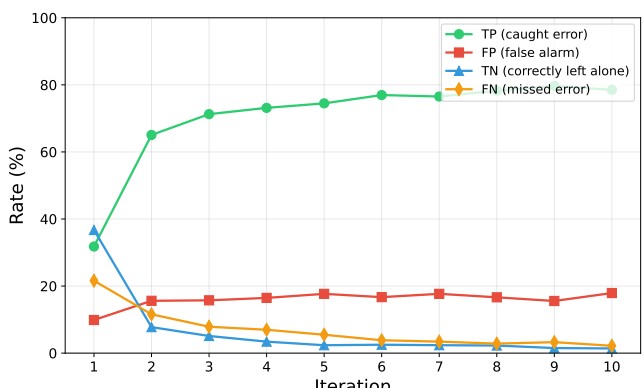

*Figure 12.* Confusion matrix components by iteration.

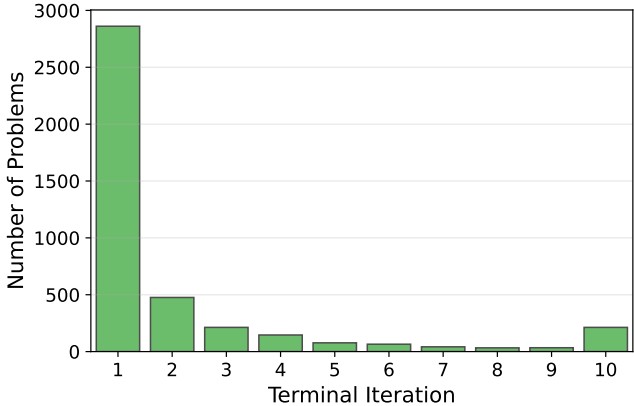

*Figure 13.* Distribution of problems by terminal iteration.

# G. Localization Analysis

**Thought-ICS oracle agreement.** Figure 14 shows the distribution of oracle localizations across three frontier models (Sonnet-3.7, GPT-4.1, GPT-5-mini). Errors concentrate in the earlier half of chains, and the oracles show strong agreement: 51% unanimous, 74% within ±1 step, 85% within ±2 steps.

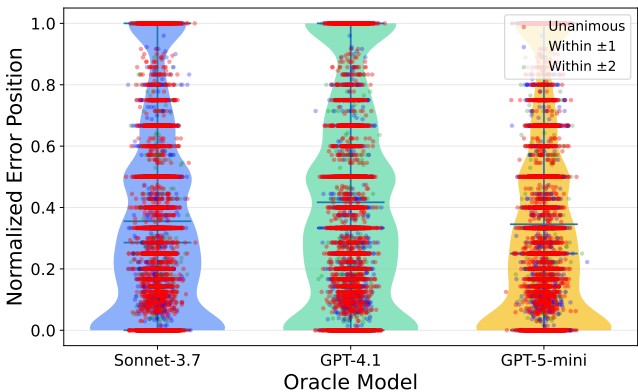

*Figure 14.* Oracle localization for Thought-ICS across three frontier models. Y-axis shows normalized error position (0 = first step, 1 = last step). Agreement: 51% unanimous (red), 74% within ±1 step (blue), 85% within ±2 steps (green).

**Robustness to localization parsing.**    The self-localization analysis in Fig. 3 uses only cleanly-parsed localizations where the model properly output the erroneous thought step in \boxed{N} format without requiring fallback heuristics (see Sec. C.4). To verify this restriction does not bias our findings, we compare clean prefix rates across different subsets of the data. Across all models, datasets, and iterations, Thought-ICS produces 18,655 self-localizations. Of these, 18,151 (97.3%) received valid oracle localizations (i.e., the frontier oracle models returned a valid step number rather than indicating no error), reflecting their high formatting instruction compliance. Separately, 8,836 (47.4%) were cleanly parsed from the evaluated models' outputs, reflecting smaller models' struggles with formatting instructions. The intersection of these two sets yields 5,133 localizations (27.5%) that satisfy both conditions; these form the cleanly-parsed localizations used in the primary analysis in Fig. 3, with fallback heuristics applied to the remaining cases. Fig. 15 compares clean prefix rates between cleanly-parsed localizations ($n = 5{,}133$) and all localizations with oracle comparisons ($n = 18{,}151$), showing minimal difference: 65.1% vs 62.6% overall, with per-model differences going in both directions. The edit accuracy gap between clean and erroneous prefixes is also consistent: +4.5% for cleanly-parsed localizations (Fig. 16) vs +5.1% for all localizations (Fig. 4), confirming the findings are robust to this filtering. All error bars represent 95% Wilson confidence intervals.

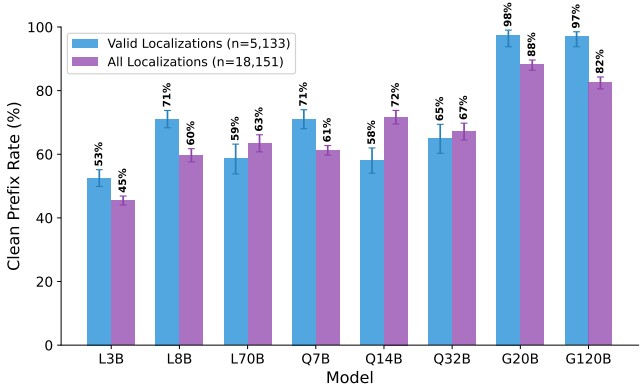

*Figure 15.* Clean prefix rate comparison between cleanly-parsed localizations and all localizations. Overall rates are similar (65.1% vs 62.6%), with per-model differences indicating no systematic bias from the filtering.

**Token-ICS localization is harder.**    Unlike Thought-ICS where models output a step number, Token-ICS requires models to quote the exact erroneous text, with truncation points found via string matching. This is fragile: minor variations in whitespace, punctuation, or paraphrasing cause failures, and only 33.5% of cases have valid positions for both self and oracle localizations (vs. 100% for Thought-ICS, where the fallback heuristics above always yield a usable step number). Among cases where all three oracles provided valid quotes ($n = 574$), only 58.2% agreed within 10% of solution length—reflecting inherent ambiguity in continuous localization (Fig. 17).

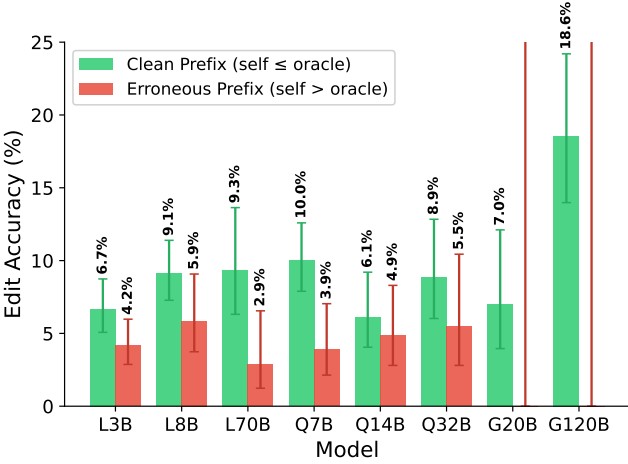

*Figure 16.* Edit accuracy by prefix type for cleanly-parsed localizations only. Clean prefixes yield higher edit accuracy (+4.5% overall), consistent with the effect observed in all localizations (+5.1%, Fig. 4).

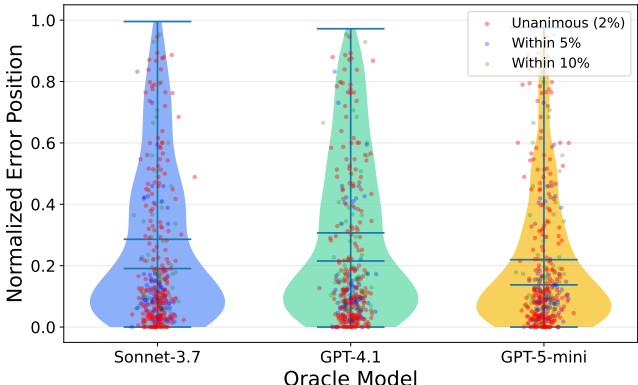

*Figure 17.* Oracle localization distribution for Token-ICS. Compared to Thought-ICS (Fig. 14), oracles show lower agreement rates, reflecting the ambiguity of continuous position localization.

**Models localize later than oracles.** Using a 10% tolerance threshold, models predominantly localize later than oracles (58.3% of cases), missing the root cause and obtaining erroneous prefixes. Clean prefix rates reach only 41.7%, compared to 60–80% for Thought-ICS (Fig. 3). Despite this, clean prefixes still yield higher edit accuracy than erroneous ones (+4.4% overall), as shown in Fig. 18, though the magnitude is smaller than Thought-ICS due to inherent noise in token-level localization.

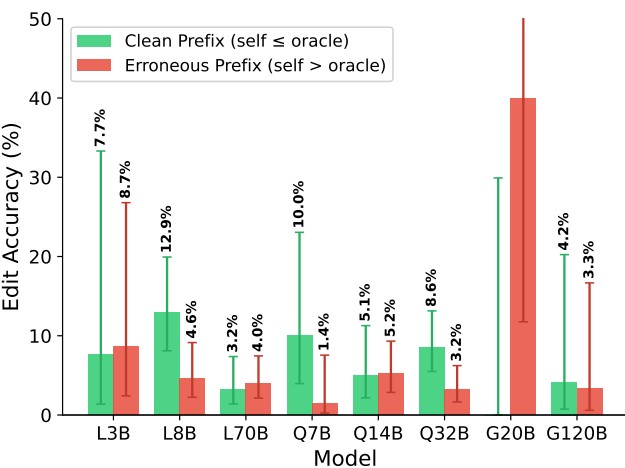

*Figure 18.* Edit accuracy by prefix type for Token-ICS. Clean prefixes show higher edit accuracy (+4.4% overall), though the effect is smaller than Thought-ICS.

## H. Oracle Verification: Full Results

Table 4 presents full results for self-correction under oracle verification across all models and datasets. The oracle verification setting provides the model with perfect error signals: the model is told exactly when its answer is wrong, but not where the error occurred. This isolates correction capability from verification noise.

*Table 4.* Self-correction under oracle verification. Init shows initial chain-of-thought accuracy. Token-ICS and Thought-ICS use oracle verification (model is told when wrong). MV@10 (majority vote over 10 samples) is included as a sampling baseline without verification. Best result per model/dataset in **bold**.

| Model | Method | AMC23 | AIME | MATH-L5 | CSQA | GPQA |
|-------|--------|-------|------|---------|------|------|
| LLaMA-3B | Init | 15.0 | 7.0 | 15.0 | 69.0 | 20.0 |
| | Token-ICS | 20.0 | 7.0 | 15.0 | 74.0 | 22.0 |
| | MV@10 | 32.5 | **28.0** | 24.0 | 75.0 | 23.0 |
| | Thought-ICS | **37.5** | 12.0 | **27.0** | **88.0** | **63.0** |
| Qwen-7B | Init | 37.5 | 16.0 | 35.0 | 71.0 | 26.0 |
| | Token-ICS | 45.0 | 20.0 | 38.0 | 76.0 | 32.0 |
| | MV@10 | 60.0 | 24.0 | 47.0 | 87.0 | 33.0 |
| | Thought-ICS | **77.5** | **32.0** | **58.0** | **94.0** | **66.0** |
| LLaMA-8B | Init | 7.5 | 4.0 | 8.0 | 70.0 | 21.0 |
| | Token-ICS | 20.0 | **18.0** | 21.0 | 76.0 | 36.0 |
| | MV@10 | 30.0 | 15.0 | 19.0 | 79.0 | 20.0 |
| | Thought-ICS | **52.5** | 15.0 | **38.0** | **93.0** | **58.0** |
| Qwen-14B | Init | 45.0 | 18.0 | 40.0 | 84.0 | 35.0 |
| | Token-ICS | 60.0 | 23.0 | 47.0 | **94.0** | 56.0 |
| | MV@10 | 72.5 | 24.0 | 53.0 | 88.0 | 38.0 |
| | Thought-ICS | **80.0** | **38.0** | **58.0** | 93.0 | **66.0** |
| OSS-20B | Init | 50.0 | 21.0 | 45.0 | 77.0 | 27.0 |
| | Token-ICS | 52.5 | 22.0 | 46.0 | 82.0 | 38.0 |
| | MV@10 | 47.5 | 20.0 | 45.0 | 80.0 | 30.0 |
| | Thought-ICS | **92.5** | **72.0** | **75.0** | **93.0** | **59.0** |
| Qwen-32B | Init | 40.0 | 18.0 | 41.0 | 89.0 | 42.0 |
| | Token-ICS | 85.0 | 35.0 | 55.0 | **96.0** | **68.0** |
| | MV@10 | 70.0 | 30.0 | 52.0 | 92.0 | 42.0 |
| | Thought-ICS | **92.5** | **62.0** | **69.0** | **96.0** | 67.0 |
| LLaMA-70B | Init | 60.0 | 39.0 | 39.0 | 85.0 | 53.0 |
| | Token-ICS | 72.5 | 47.0 | 49.0 | 90.0 | 69.0 |
| | MV@10 | 62.5 | 52.0 | 48.0 | 89.0 | 59.0 |
| | Thought-ICS | **77.5** | **54.0** | **62.0** | **94.0** | **73.0** |
| OSS-120B | Init | 57.5 | 24.0 | 42.0 | 84.0 | 45.0 |
| | Token-ICS | 72.5 | 33.0 | 47.0 | 92.0 | 56.0 |
| | MV@10 | 57.5 | 24.0 | 28.0 | 90.0 | 43.0 |
| | Thought-ICS | **92.5** | **70.0** | **73.0** | **94.0** | **79.0** |

## I. Compute Analysis: Token-ICS vs Thought-ICS

We compare per-problem compute between Token-ICS and Thought-ICS on the oracle-verification runs from Table 4, excluding the shared initial CoT. For each model and dataset, we replay the saved correction traces and count new output tokens generated during the correction phase: localization outputs plus regenerated solution tokens past the kept prefix. Tokens are counted with each model's native HuggingFace tokenizer. We also report prefix saved, the fraction of resampling tokens skipped via the KV-cached prefix.

Averaged over the evaluated datasets (Table 5), Thought-ICS uses 2,610 new tokens vs. Token-ICS's 2,150 ($1.21\times$) but achieves 19.1 percentage points higher accuracy (67.2% vs. 48.1%), a 13% reduction in tokens per correct answer.

The aggregate is dominated by small and mid-scale models, where Thought-ICS pays a token premium for large accuracy gains. The pattern inverts at the frontier: Qwen-32B uses 38% fewer correction tokens at 9.7 percentage points higher accuracy, and OSS-120B uses 46% fewer at 22.4 percentage points higher. Thought-ICS generates fewer new tokens per iteration on these models, despite running more iterations.

The prefix-saved column isolates the contribution of KV-cached backtracking. Thought-ICS saves 30% of would-be resampling tokens vs. 16% for Token-ICS. Token-ICS localization requires the model to emit an exact substring quote that string-matches against its prior solution; matches fail frequently (Appendix G), forcing regeneration from scratch.

*Table 5.* Compute per problem on oracle-verification runs, averaged over the evaluated datasets. New Tok.: average new output tokens generated per problem during correction (localization output plus regenerated solution past the kept prefix); initial CoT excluded. Acc.%: oracle-verification accuracy from Table 4. Saved: fraction of resampling tokens skipped via KV-cached prefix. Tokens use each model's native HuggingFace tokenizer.

| | Token-ICS | | | Thought-ICS | | |
|---|---|---|---|---|---|---|
| Model | New Tok. | Acc.% | Saved | New Tok. | Acc.% | Saved |
| LLaMA-3B | 253 | 27.6 | 16% | 2,904 | 47.0 | 42% |
| Qwen-7B | 1,296 | 42.2 | 14% | 3,397 | 66.1 | 37% |
| LLaMA-8B | 3,395 | 34.2 | 9% | 3,645 | 52.3 | 29% |
| Qwen-14B | 2,588 | 56.2 | 24% | 2,953 | 67.4 | 28% |
| OSS-20B | 1,532 | 48.1 | 6% | 1,885 | 77.1 | 25% |
| Qwen-32B | 3,171 | 68.2 | 38% | 1,954 | 77.9 | 28% |
| OSS-120B | 2,816 | 60.1 | 6% | 1,533 | 82.5 | 22% |
| **Average** | 2,150 | 48.1 | 16% | 2,610 | 67.2 | 30% |

## J. Correction Diversity

We analyze the diversity of answers produced across correction iterations on the oracle-verification runs, restricting to problems with more than one correction iteration (Table 6).

77.5% of Thought-ICS correction iterations change the answer relative to the initial attempt, vs. 58.9% for Token-ICS. Consecutive iterations under Token-ICS produce the same answer 44.6% of the time vs. 30.6% for Thought-ICS, with the gap widest on the largest OSS models (76–79% vs. 50–55%). The Token-ICS unique-answer ratio is higher (0.65 vs. 0.57), but this reflects Token-ICS's fewer self-correction attempts (4.3 vs. 8.5 iterations on the same subset) rather than greater diversity per step; the per-step metrics consistently favor Thought-ICS.

*Table 6.* Correction diversity on problems with more than one correction iteration. #Probs: number of such problems. Avg Iters: average total iterations on those problems. Unique Ans.: ratio of unique answers to total iterations (1.0 = every iteration produces a different answer). Consec. Same: fraction of consecutive iteration pairs with the identical answer. Changed: fraction of correction iterations whose answer differs from the initial attempt.

| Model | Method | #Probs | Avg Iters | Unique Ans. | Consec. Same | Changed |
|---|---|---|---|---|---|---|
| LLaMA-3B | Token-ICS | 65 | 2.7 | 0.77 | 36.2% | 62.6% |
| | Thought-ICS | 338 | 8.9 | 0.58 | 26.1% | 83.1% |
| Qwen-7B | Token-ICS | 127 | 3.1 | 0.76 | 34.4% | 66.6% |
| | Thought-ICS | 257 | 8.5 | 0.61 | 22.3% | 78.0% |
| LLaMA-8B | Token-ICS | 219 | 3.7 | 0.79 | 23.9% | 80.4% |
| | Thought-ICS | 323 | 8.6 | 0.60 | 11.6% | 80.1% |
| Qwen-14B | Token-ICS | 170 | 4.6 | 0.63 | 45.2% | 60.3% |
| | Thought-ICS | 229 | 8.5 | 0.61 | 24.0% | 77.5% |
| OSS-20B | Token-ICS | 118 | 3.3 | 0.48 | 78.8% | 20.0% |
| | Thought-ICS | 165 | 8.5 | 0.43 | 54.5% | 71.0% |
| Qwen-32B | Token-ICS | 213 | 5.9 | 0.67 | 32.8% | 76.8% |
| | Thought-ICS | 173 | 8.2 | 0.57 | 30.0% | 72.4% |
| OSS-120B | Token-ICS | 152 | 4.7 | 0.43 | 76.1% | 23.5% |
| | Thought-ICS | 146 | 7.7 | 0.48 | 50.1% | 70.9% |
| **Average** | Token-ICS | 1,064 | 4.3 | 0.65 | 44.6% | 58.9% |
| | Thought-ICS | 1,631 | 8.5 | 0.57 | 30.6% | 77.5% |

## K. Chain and Thought Length Distributions

Figure 19 shows chain and thought length distributions by model. Larger models tend to be more verbose; chain lengths range from 5.8 steps (Qwen-7B) to 21.0 steps (Qwen-32B), and thought lengths range from 34 tokens (LLaMA-3B) to 186 tokens (OSS-20B). Figure 20 shows that incorrect traces are longer on average (12.6 steps) than correct traces (10.0 steps).

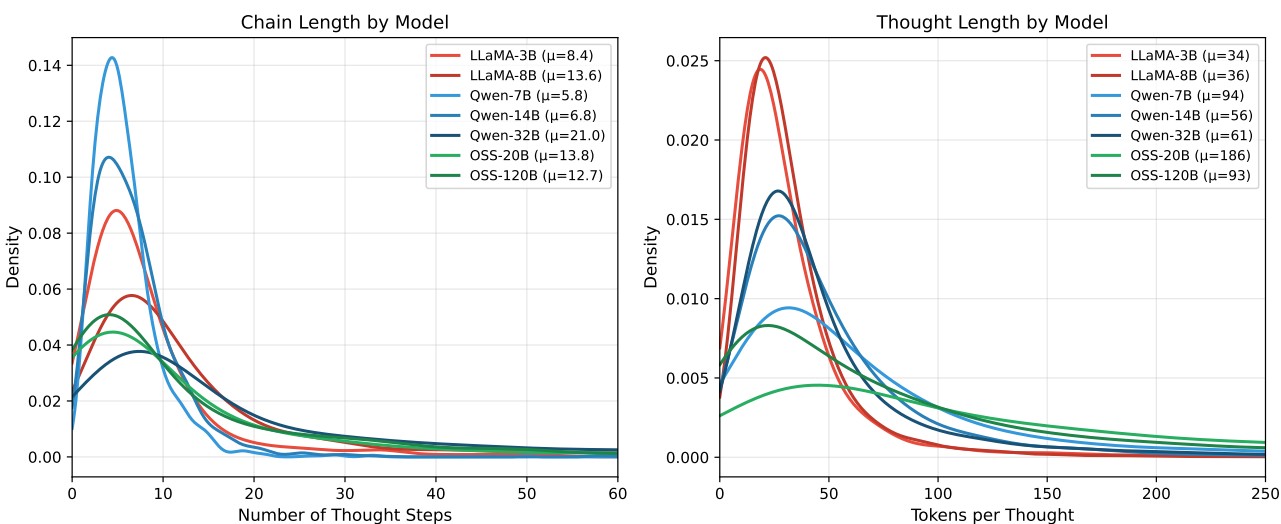

*Figure 19.* Chain length (left) and thought length (right) distributions by model.

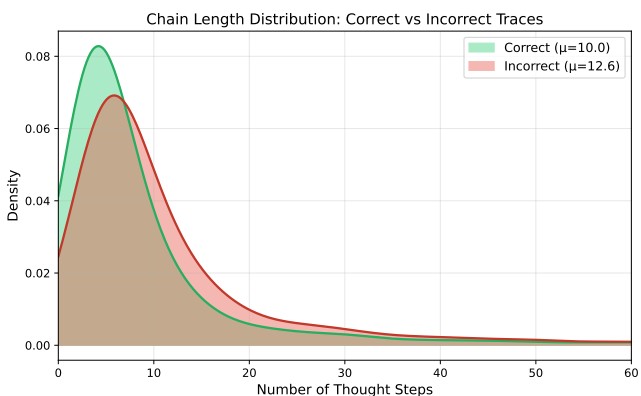

*Figure 20.* Chain length distribution by correctness.

## L. Fully Autonomous Self-Correction: Full Results

Table 7 presents full results for Thought-ICS compared to Self-Refine and CoVe across all eight models (3B–120B). Models in the 14B–120B range show consistent gains from Thought-ICS-A. For smaller models (3B–8B), CoT often remains the best result, indicating that self-correction can hurt weaker models.

*Table 7.* Full results comparing self-correction methods across all models. CoT shows initial chain-of-thought accuracy. Self-Refine and CoVe are unstructured methods; Thought-ICS is our structured method with two variants: Thought-ICS-S (self-verification only) and Thought-ICS-A (with confidence safeguard that resets to initial answer on low-confidence exits; see Sec. 6). Best result per model/dataset in **bold**.

| Model | Method | AMC23 | AIME | MATH-L5 | CSQA | GPQA | MathQA |
|---|---|---|---|---|---|---|---|
| LLaMA-3B | CoT | **15.0** | 7.0 | **15.0** | **69.0** | 20.0 | **48.0** |
| | Self-Refine | 7.5 | **9.0** | 14.0 | 62.0 | 17.0 | 35.0 |
| | CoVe | 12.5 | 8.0 | 14.0 | **69.0** | 20.0 | 38.0 |
| | Thought-ICS-S | **15.0** | 5.0 | 9.0 | 56.0 | 28.0 | 44.0 |
| | Thought-ICS-A | **15.0** | 5.0 | 9.0 | 56.0 | **29.0** | 44.0 |
| Qwen-7B | CoT | 37.5 | 16.0 | 35.0 | 71.0 | 26.0 | 69.0 |
| | Self-Refine | **45.0** | **21.0** | 38.0 | 73.0 | **31.0** | 70.0 |
| | CoVe | 42.5 | 18.0 | 31.0 | 67.0 | 26.0 | 64.0 |
| | Thought-ICS-S | 40.0 | 16.0 | **40.0** | 85.0 | 28.0 | **71.0** |
| | Thought-ICS-A | 40.0 | 16.0 | **40.0** | 85.0 | 28.0 | **71.0** |
| LLaMA-8B | CoT | 7.5 | **4.0** | 8.0 | **70.0** | 21.0 | **57.0** |
| | Self-Refine | 7.5 | **4.0** | 9.0 | 69.0 | **26.0** | 55.0 |
| | CoVe | 7.5 | 3.0 | 8.0 | **70.0** | 18.0 | 39.0 |
| | Thought-ICS-S | **22.5** | 3.0 | **15.0** | 67.0 | 25.0 | 53.0 |
| | Thought-ICS-A | **22.5** | 3.0 | **15.0** | 67.0 | 25.0 | 53.0 |
| Qwen-14B | CoT | 45.0 | 18.0 | 40.0 | 84.0 | 35.0 | **84.0** |
| | Self-Refine | 40.0 | 19.0 | 40.0 | 85.0 | 33.0 | 79.0 |
| | CoVe | 45.0 | 17.0 | 37.0 | 76.0 | **38.0** | 77.0 |
| | Thought-ICS-S | **50.0** | **20.0** | **43.0** | 90.0 | **38.0** | 83.0 |
| | Thought-ICS-A | **50.0** | **20.0** | **43.0** | 90.0 | **38.0** | 83.0 |
| OSS-20B | CoT | 50.0 | 21.0 | 45.0 | 77.0 | 27.0 | 71.0 |
| | Self-Refine | 52.5 | 26.0 | 52.0 | 78.0 | 31.0 | 73.0 |
| | CoVe | 57.5 | 39.0 | 53.0 | 74.0 | 35.0 | 53.0 |
| | Thought-ICS-S | **80.0** | 30.0 | 52.0 | **81.0** | 46.0 | 80.0 |
| | Thought-ICS-A | **80.0** | 63.0 | 59.0 | **81.0** | 49.0 | 82.0 |
| Qwen-32B | CoT | 40.0 | 18.0 | 41.0 | 89.0 | 42.0 | **86.0** |
| | Self-Refine | 65.0 | 27.0 | 47.0 | 88.0 | 39.0 | 84.0 |
| | CoVe | **72.5** | 31.0 | **56.0** | 86.0 | 39.0 | 79.0 |
| | Thought-ICS-S | 65.0 | **47.0** | **56.0** | 90.0 | **43.0** | 85.0 |
| | Thought-ICS-A | 67.5 | **47.0** | **56.0** | 90.0 | **43.0** | **86.0** |
| LLaMA-70B | CoT | 60.0 | 39.0 | **39.0** | 85.0 | 53.0 | 73.0 |
| | Self-Refine | 60.0 | 44.0 | 35.0 | 84.0 | 51.0 | **76.0** |
| | CoVe | 55.0 | **45.0** | 37.0 | 78.0 | 46.0 | 69.0 |
| | Thought-ICS-S | **65.0** | 38.0 | 38.0 | **87.0** | 62.0 | 74.0 |
| | Thought-ICS-A | 62.5 | 42.0 | 38.0 | **87.0** | **63.0** | **76.0** |
| OSS-120B | CoT | 57.5 | 24.0 | 42.0 | 84.0 | 45.0 | 79.0 |
| | Self-Refine | 60.0 | 26.0 | 44.0 | 80.0 | 45.0 | 79.0 |
| | CoVe | 67.5 | 36.0 | 47.0 | **85.0** | 52.0 | 84.0 |
| | Thought-ICS-S | 75.0 | 46.0 | 52.0 | 82.0 | 64.0 | 87.0 |
| | Thought-ICS-A | **82.5** | **67.0** | **57.0** | 82.0 | **69.0** | **90.0** |

