# OpenReview forum: "Structure Enables Effective Self-Localization of Errors in LLMs"
_ICML.cc/2026/Conference — ICML 2026 regular_

### Official Review · Reviewer_iVZh · 2026-03-10

**Soundness:** 2
**Presentation:** 3
**Significance:** 2
**Originality:** 3
**Overall Recommendation:** 3
**Confidence:** 3

**Summary:**

The paper studies the self-correction ability of LLMs and points out that the traditional unstructured thought chain (CoT) reasoning process hinders the model's ability to identify its own errors. To address this issue, the author proposes the Thought-ICS framework. This framework models the reasoning process as a thought Markov Decision Process (MDP). By guiding the model to generate a discrete thought at a time, this framework can create natural boundaries, thereby locating and correcting errors. Additionally, the author discovers the phenomenon of specificity collapse, where the model mistakenly labels the correct answer as incorrect in multiple iterations, which is the main bottleneck for the model's self-validation. By introducing a confidence guarantee mechanism, reverting to the initial answer when the confidence is low, this system achieves a net positive error correction improvement and outperforms baseline methods such as Self-Refine and CoVe.

**Compliance With Llm Reviewing Policy:**

Affirmed.

**Final Justification:**

My core concern has only been partially addressed: the method’s practical gains seem to stem more from structured prompting and the backtracking scaffold than from any substantive methodological innovation introduced by the MDP formalism itself. Therefore, I am keeping my original rating unchanged.

**Key Questions For Authors:**

1. In Appendix G, you sincerely stated that the precise string matching extraction method of Token-ICS is "fragile", which also led to its oracle consistency being only 58%. Could this vulnerability in the evaluation metrics affect the fairness of the performance comparison between Thought-ICS and Token-ICS? Have you considered using more flexible metrics when evaluating the localization capability of Token-ICS?

2. The paper introduced the rigorous formal terms (such as state space, action space, and transition function) from the Markov Decision Process (MDP) to define Thought-ICS. However, from the implementation details in Appendix C.1, it seems that this method is mainly a set of very pure and ingenious prompt engineering combined with string truncation control, without involving any modification to the underlying decoding algorithm of the model or training of the value function. What are the specific advantages and necessity of using this MDP theoretical framework for definition? Could it be considered to more clearly define the theoretical boundaries of this method in the main text?

3. Just as you sincerely pointed out in the section on Limitations, the current assessment focuses on closed-ended reasoning benchmarks with verifiable answers. Given this limitation, I would like to further explore the theoretical generalizability boundaries of this method: Have you conducted any preliminary observations or tests of this method in open-ended generation tasks (such as code generation, long-form writing)? In these tasks, the boundaries of the reasoning steps are often less clear, the definitions of errors are more subjective, and the verification methods (such as code execution feedback) also significantly differ from those of closed-ended question answering. May I ask you, do you think the discrete structure of Thought MDP, when faced with continuous semantic flows or more complex verification scenarios, can still maintain its structured advantages, or will it disrupt the original coherence of the model's thinking?

**Limitations:**

yes

**Strengths And Weaknesses:**

Soundness: The experimental design is rigorous. The evaluation process is divided into two parts: oracle verification and autonomous self-verification. The positioning and error-correction capabilities of the model are decoupled from its verification capabilities. The finding that the probability of correct answers decreases as the number of iterations increases has high empirical value. This finding explains many reasons for the failure of autonomous self-error-correction loops.
When there is a discrepancy in V/L or when the maximum number of iterations is reached, Thought-ICS-A will reset to the response of iteration 0. Although this reset mechanism is a very practical engineering safeguard measure, it is mainly an avoidance mechanism rather than a fundamental enhancement of the model's verification and correction of its own reasoning capabilities.

Presentation:  The article has a clear structure and a smooth writing style. Figure 1 clearly and intuitively presents the backtracking mechanism of Thought-ICS. The narrative logic is rigorous, starting from identifying the structural issues in the CoT, gradually delving into verifying the model's capabilities under the oracle condition, and ultimately exploring the practical challenges in autonomous deployment.

Significance: LLM self-correction is a key topic in the field of reasoning research. The proposed structural framework in this paper does not require external rewards and provides practitioners with meaningful guidance without the need for specialized training models. However, current evaluations mainly focus on verifiable closed-set reasoning tasks. If this method can be extended to open domains or tasks with execution feedback (such as programming and writing), it will be more meaningful.

Originality: Applying MDP to the self-generated semantic boundaries of LLMs for achieving retroactive self-localization is a clever and novel approach. It provides a new perspective for the overall method of word-by-word analysis.

---

> ### Author Rebuttal · Authors · 2026-03-31
>
> We thank the reviewer for their thoughtful and detailed evaluation.
>
> **W1: Reset mechanism is avoidance, not fundamental improvement of verification/correction.**
>
> We agree the confidence safeguard is not a fundamental enhancement of verification—it is
> a principled response to the specificity collapse phenomenon we identify. The key
> contribution is the *diagnosis*: we show that autonomous self-correction degrades
> primarily because the model's verifier incorrectly rejects correct answers over iterations,
> not because localization or resampling fail. The safeguard is one practical mitigation, but
> the finding itself—that verification specificity is the bottleneck, not correction
> capability—provides actionable guidance for future work (e.g., training better verifiers,
> calibrating confidence). We note that under oracle verification, Thought-ICS achieves strong
> positive correction without any safeguard, confirming that the correction mechanism itself
> is sound.
>
> **W2 \& Q3: Limited to closed-ended tasks; generalizability to open-ended tasks.**
>
> We note that our evaluation is not limited to step-by-step mathematical reasoning—we also
> test on scientific reasoning (GPQA) and commonsense reasoning (CSQA), where Thought-ICS is
> effective. We do not believe code generation is open-ended, as it has objective verifiers via test execution and environmental feedback. We consider truly non-verifiable settings (e.g., long-form writing) as future
> work and out of scope, since verification is a fundamentally different paradigm in the
> absence of objective truth.
>
> **Q1: Token-ICS string matching is "fragile" (58% oracle consistency). Does this unfairly disadvantage Token-ICS?**
>
> We took steps to ensure fairness: Token-ICS oracle agreement is measured with a relaxed
> 10% tolerance threshold (positions matching within 10% of solution length), and when
> quote matching fails at runtime, Token-ICS falls back to full regeneration rather than
> producing a corrupted prefix. The lower agreement reflects inherent ambiguity in continuous
> localization—there is no single "correct" character position to truncate at, unlike
> step-level localization where the error boundary is well-defined.
>
> **Q2: MDP formalism vs implementation gap.**
>
> The MDP formalism formalizes reasoning as sampling a series of successive reasoning steps
> rather than a single monolithic block. This provides a precise vocabulary for the design
> space: states are verified prefixes, actions are next thoughts, and transitions extend or
> backtrack—clarifying how alternatives compare (e.g., Token-ICS vs Thought-ICS differ in
> state representation, not algorithm). It also connects prompt-based inference-time
> correction to the RL literature, where the same MDP can be optimized via learned value
> functions or reward shaping. Our implementation is deliberately prompt-based to isolate the
> effect of structured decomposition without confounding it with training. We will clarify
> this motivation in the revision.
>
> Additionally, in response to other reviewers' feedback, we have added a compute analysis
> comparing token usage across all methods (Token-ICS, Thought-ICS, CoVe, Self-Refine) in
> both oracle and autonomous settings, as well as a resampling variability analysis. These
> further demonstrate the practical efficiency of Thought-ICS—see our response to Reviewer
> DqUW for details.
>
> We appreciate the reviewer's careful reading and insightful questions, particularly
> regarding the MDP formalism and Token-ICS fairness. We hope these responses clarify our
> design choices and would be grateful if the reviewer would consider increasing their score.

---

> > ### Author Rebuttal · Reviewer_iVZh · 2026-04-03
> >
> > Thank you for your  response. One concern I still have is that the paper’s claimed structured advantage seems to rest heavily on a formal MDP framing, while the actual method appears in practice to be mostly prompt engineering plus truncation control.Therefore, I will maintain my original rating.

---

> > > ### Author Response · Authors · 2026-04-05
> > >
> > > We respectfully disagree with the characterization that this work is "mostly prompt engineering plus truncation control."
> > >
> > > **Generally speaking, inference-time self-correction methods operate through prompting.** Consider the related work that relies on purely self-guided or binary feedback signals:
> > >
> > > - **Self-Refine** (Madaan et al., 2023) prompts the model to critique and regenerate its own response
> > > - **CoVe** (Dhuliawala et al., 2023) derives verification questions from the chain-of-thought to guide revision
> > > - **Reflexion** (Shinn et al., 2023) uses task outcome signals to generate verbal reflections stored in episodic memory
> > > - **RCI** (Kim et al., 2023) uses environment feedback for computer tasks in a recursive critique loop
> > >
> > > These are all published, well-cited works that have meaningfully advanced the field, and all operate through prompting. We do not believe that using prompts as the interface to the LLM diminishes these contributions, nor ours.
> > >
> > > **What distinguishes Thought-ICS is the pipeline design.** Our system decomposes self-correction into its constituent capabilities — *verification*, *localization*, and *resampling* — and operates at thought-level granularity rather than treating reasoning traces as monolithic. This decomposition allows us to:
> > >
> > > 1. **Isolate and study each capability independently** — something no other paper in this area does at our scale. We show that localization and resampling work well, but *verification* is the bottleneck — a finding (specificity collapse) that was only possible because of our structured decomposition.
> > > 2. **Perform targeted correction** — localizing the *first* erroneous thought and resampling from the verified-correct prefix, rather than regenerating the entire response from holistic critique. This yields a 77.5% answer-change rate vs. 58.9% for token-level approaches, demonstrating that the structure produces materially different and better corrections.
> > >
> > > **Our work overturns a key finding in the literature.** As we note in our related work: "Huang et al. (2024) and Stechly et al. (2024) show that without external signals, models struggle to verify their own reasoning and often degrade performance through self-correction. Tyen et al. (2024) decomposes self-correction into localization and correction, finding that LLMs cannot self-localize errors effectively, but can correct when given error locations by an oracle. We find that with appropriate structure, this limitation can be overcome." We are the first to demonstrate that language models *can* localize errors in their own reasoning in a manner that enables reliable correction — and that thought-level structure is the key enabler. This is a substantive finding that challenges the prevailing understanding in this area.
> > >
> > > **On the scope of our empirical study:** No other self-correction paper studies the range of models that we do — 8 models from 3B to 120B parameters across 6 benchmarks, under both oracle and autonomous verification. This evaluation is what enabled our diagnostic findings about how self-correction behavior varies with model scale — challenges, findings, and insights that were previously unexplored and are important to understanding how to design systems that effectively self-correct at inference time.
> > >
> > > **On the MDP framing:** To be clear, the MDP formalism is *not* our claimed contribution — it provides conceptual grounding to situate the design space. The contributions are the pipeline itself, its empirical advantages (20–40% correction lift under oracle verification; superior token efficiency over Self-Refine and CoVe in autonomous mode), and the diagnostic findings that emerged from studying it.
> > >
> > > We believe the contribution of a method should be judged by the system it builds, the findings it produces, and the empirical rigor of its evaluation — not solely by whether it uses prompts as an interface. The behaviors our structured approach elicits do not surface nearly as consistently or effectively in less-structured attempts (i.e., the baseline methods we compare against), and understanding why is a meaningful research contribution. We respectfully ask the reviewer to reconsider their assessment.

---

### Official Review · Reviewer_ex6G · 2026-03-13

**Soundness:** 3
**Presentation:** 3
**Significance:** 2
**Originality:** 3
**Overall Recommendation:** 4
**Confidence:** 4

**Summary:**

This paper proposes Thought-ICS, a framework for automated self-correction. The framework consists of three steps; 1) verification or errors, 2) localisation, and 3) Re-sampling (re-generation of thoughts). The experimental results show that the proposed structured generation and self-correction outperforms token-level ICS in terms of localising error, and precise localisation yields more precise resampling, resulting in accurate self-correction. The authors also investigate the practical limitation of the proposed approach in terms of building a fully autonomous self-correction system and propose a remedy.

**Compliance With Llm Reviewing Policy:**

Affirmed.

**Final Justification:**

The rebuttal addressed my main concerns, but it does not significantly affect my decision.

Overall, the paper is well written, has a clear structure, and addresses an important topic; however, it lacks novelty from a methodological perspective.

In the final response from the authors, they state "If relying on prompting limits novelty, this applies equally to the entire subfield." and "there key finding is novel". Actually I agree to both statements. It is true that relying on prompting limits novelty and it applies equally to the entire subfield. However, this does not mean that the work is not worth publishing, as novelty is not the only criterion for publication. Even if the method is not particularly novel, the paper could still be a good publication if it presents meaningful and important findings.

If I believed the paper’s findings were not novel, I would likely lean toward rejection. However, I do agree that the authors’ findings are novel, although they lack practical novelty. This is the main reason I gave a weak accept rating (a technically solid paper that advances at least one sub-area of AI, with a contribution that others are likely to build on, but with some weaknesses that limit its impact).

As stated in my rebuttal response, if the paper were submitted to an industry track, where practicality and applicability are prioritised, the lack of methodological novelty would be less of a concern, and I would lean toward 5: Accept. However, in a regular track, where methodological novelty is also an important criterion, this limitation becomes a clear weakness.

In summary, I lean toward acceptance, but the paper has weaknesses that limit its impact in a regular track context.

**Key Questions For Authors:**

1. While Thought-ICS appears to outperform Token-ICS, it seems that it may also require higher token consumption. Have the authors compared the token usage between the two approaches?

2. How were the ground-truth labels obtained to compute recall, specificity, and accuracy in the experiments reported in Section 6?

3. In Figure 8, the performance difference between the settings with and without the confidence safeguard appears marginal when the number of iterations ranges from 2 to 6. Have the authors conducted statistical significance tests to verify whether this performance gap is meaningful?

**Limitations:**

Yes

**Strengths And Weaknesses:**

***Strengths**

1. The authors conducted extensive experiments across diverse models an in-depth analysis.

2. The proposed Thought-ICS approach is intuitive and easy to reproduce. The approach itself is a simple workflow but the results show that it is simple-but-highly effective and no training process is required. The method is highly scalable and has potential to be applied across diverse industry sectors.

3. The paper is well-written with easy to understand languages, and neat and intuitive graphical results are provided.

**Weakness**

1. As noted by the authors, the self-verification process is a key component of the proposed framework. However, different use cases may have different definitions of correctness. For instance, some applications may tolerate small rounding errors or reasonable assumptions, even if they are not explicitly stated in the given context, whereas others are highly regulated and do not allow any errors. Furthermore, since the self-verification process is performed through LLM prompting, it may exhibit judgment biases that differ from those of the actual stakeholders or use-case owners (10.18653/v1/2024.findings-acl.29, 10.18653/v1/2024.emnlp-main.474, 10.18653/v1/2025.emnlp-industry.82). The proposed approach does not include components designed to address these issues.

2. Although the design of the framework may be novel, it's a simple chain-of-LLM prompting, without additional use of tools or auxiliary features (e.g., the components that can address above issues), which reduces the significance of the work.

3. As an autonomous correction method, the proposed approach is compared only with Self-Refine and CoVe, both of which were published in 2023 and are relatively outdated. Comparisons with more recent methods (e.g., https://openreview.net/forum?id=CjwERcAU7w) would be necessary to better verify the advantages of the proposed approach.

4. The paper appears somewhat redundant in several places. For example, Algorithms 1 and 2 are already clearly described in the main text, making the inclusion of pseudocode somewhat unnecessary. In addition, Lines 227–234 largely repeat the content of Section 4, and Lines 373–376 (“This step gates the correct loop: …”) also repeat material from Section 4 (Verification). Removing these repetitions and instead including Appendix E (implementation details for the experiments in Section 6.3) would likely improve the clarity and overall quality of the paper.

5. Minor errors:
- The manuscript does not define the abbreviation “MDP”.
- Line 199 refers to “Algorithm 2”, although no such algorithm appears in the manuscript. The corresponding item seems to be “Framework 2”.

---

> ### Author Rebuttal · Authors · 2026-03-31
>
> We thank the reviewer for their detailed and constructive review.
>
> **W1: Self-verification bias — different use cases have different correctness definitions; LLM prompting may exhibit judgment biases.**
>
> This is precisely why we structure our paper to first study correction under oracle
> verification, isolating the model's localization and resampling capabilities from
> verification biases. We then dedicate Section 6 to analyzing autonomous self-verification,
> where we identify specificity collapse—the model incorrectly rejecting correct answers—as
> the dominant failure mode. The reviewer's point about domain-specific correctness definitions
> is well-taken; our framework is modular, and the verification component can be replaced with
> domain-appropriate verifiers (e.g., execution feedback for code, domain-expert review)
> without changing the localization and resampling pipeline.
>
> **W2: Simple chain-of-LLM prompting without tools or auxiliary features reduces significance.**
>
> We view the simplicity as a strength: by using only prompting without tools or training, we
> isolate the effect of structured decomposition on self-correction. Our contribution is not a
> complex system but a thorough empirical analysis across 8 models and 6 benchmarks that
> identifies when and why self-correction works or fails. The findings (e.g., specificity collapse, the effectiveness of step-level backtracking, the
> importance of discrete reasoning structure for localization) are applicable regardless of
> whether future systems add tools or training on top.
>
> **W3: Outdated baselines — only Self-Refine and CoVe (2023). Comparison with more recent methods needed.**
>
> Our scope is specifically self-correction *without external feedback or with only binary
> correctness signals*—a deliberately constrained setting. As discussed in our related work,
> very few methods operate under these conditions; Self-Refine and CoVe are the primary ones
> (we exclude methods that rely on environmental feedback such as code execution, which is a
> different setting). The cited paper SCoRe trains self-corrective ability, and itself acknowledges that
> inference-only self-correction does not work without their training procedure. Our work is
> complementary: we conduct a thorough study of what works and what does not *purely at
> inference time*, isolating failure modes (e.g., specificity collapse) and addressing them
> through inference-time adjustments *without any additional training*. These findings
> can inform training-based approaches like SCoRe.
>
> **W4: Redundancy in presentation (Algorithms 1/2, Lines 227--234, Lines 373--376).**
>
> Thank you—we will remove the redundant passages and move Appendix E into the main paper
> in the revision.
>
> **W5: Minor errors — MDP not defined; Line 199 "Algorithm 2" should be "Framework 2".**
>
> Thank you for catching these. We will define MDP on first use and fix the reference in the
> revision.
>
> **Q1: Have the authors compared token usage between Thought-ICS and Token-ICS?**
>
> Yes. Under oracle verification, Thought-ICS uses $1.16\times$ more correction tokens than
> Token-ICS ($2{,}039$ vs $1{,}750$ avg per problem) but achieves $+19.1$ pp higher accuracy,
> making it $1.2\times$ more token-efficient per accuracy point. In the autonomous setting,
> Thought-ICS-A uses the fewest correction tokens ($1{,}133$) while achieving the highest
> accuracy ($62.9%$) compared to CoVe ($2{,}592$ tokens, $46.3%$) and Self-Refine ($1{,}415$
> tokens, $46.6%$).
>
> **Q2: How were ground-truth labels obtained for recall, specificity, accuracy in Section 6?**
>
> The ground-truth labels come from the datasets themselves—each benchmark provides
> reference answers. We use these to determine whether a given iteration's answer is correct,
> which in turn defines whether the verifier's accept/reject decision was a true/false
> positive/negative for computing recall and specificity.
>
> **Q3: Figure 8 confidence safeguard — performance gap seems marginal for iterations 2--6. Statistical significance tests?**
>
> We ran McNemar's test (paired, per-problem) comparing correctness with vs without the
> safeguard at each terminal iteration. The safeguard is statistically significant at
> iterations 2 ($p{=}0.018$), 3 ($p{=}0.014$), 4 ($p{=}0.003$), and 10 ($p{<}0.001$).
> Iterations 5--9 do not individually reach significance due to smaller sample sizes
> ($n{=}33$--$77$), but show consistent improvements of $+6.2$ to $+12.1$ pp.
>
> We hope the new statistical significance tests, token efficiency analysis, and
> clarifications on baselines and verification address the reviewer's concerns. We will fix
> the presentation issues (redundancy, minor errors) in the revision and would be grateful if
> the reviewer would consider increasing their score.

---

> > ### Author Rebuttal · Reviewer_ex6G · 2026-04-02
> >
> > Thank you for clarifying my questions. It greatly helped to better understand the paper. Please find my additional comments below:
> >
> > **W3**: I see that the authors focused their comparisons on self-correction without external feedback or with only binary correctness signals, since these are the most direct benchmarks for their proposed approach. Therefore, I’d like to revise my assessment of this weakness to a suggestion. Even if the authors restrict their scope to *self-correction without external feedback or with only binary correctness signals*, it would still be valuable to include self-correction methods that go beyond this scope. If the proposed method outperforms more advanced approaches, such as those using LLM feedback, it would make for an outstanding publication. Even if it performs slightly worse than these complex methods, it still offers practical advantages in terms of computational efficiency and cost, provided the performance is comparable. In either case, the results strengthen the paper’s claims, making the work achieve higher applicability.
> >
> > Overall, I have decided to keep my current score. This was a difficult decision, so I’d like to clarify my rationale of judgment. I do agree with the authors’ statement in their W2 response—*We view the simplicity as a strength*—since simplicity greatly enhances applicability. In fact, this aligns closely with my own research philosophy. But on the other hand, it is also true that simplicity and broad applicability do not necessarily equate to high novelty. However applicable and practical the approach may be, the method itself is essentially a sequence of LLM prompting, as the authors themselves acknowledge. Therefore, my original decision was made, based on the following core considerations [3-scale: Good, Neutral, Bad]:
> >
> > - **Work’s implications**: Good (the paper presents solid findings supported by extensive experiments)
> > - **Applicability and Practicality**: Neutral (mainly due to the limited comparison with baselines, as mentioned above)
> > - **Novelty**: Bad
> >
> > This led me to a conclusion of Weak accept: **Technically solid paper that advances at least one sub-area of AI**, with a contribution that others are likely to build on, **but with some weaknesses that limit its impact (e.g., limited evaluation)**.
> >
> > With the rebuttal, although several concerns have been resolved, my evaluation on the three core dimensions are the same.
> > If the paper were to include comparisons with other self-correction methods over the defined scope, it would significantly enhance the practicality of the proposed approach, as mentioned above, and could sufficiently offset the weakness in *Novelty*. In that case, I must admit that I would lean toward a score of `5: Accept`. Or, if the paper is intended for an Industry-Track submission—such as those in ACL-related conferences, which likely to emphasise applicability and practicality over novelty—I would also inclined to `5: Accept`. However, for a normal paper track, novelty is also important as practicality. Therefore, I've decided to keep my previous score.
> >
> > Nonetheless, to the best of my knowledge, ICML 2026 does not have Industry Track submissions; it only accepts submissions to Regular Track and Position Paper Track. **Hence, although I'm giving an overall score of 4 through this review based on the conventional definition of Regular Track, I would like to leave my comment to the Area Chair that if ICML 2026 intends to accommodate both the conventional regular track and industry track papers, my overall score is 5.**
> >
> > I hope I've made the explanation sufficiently clear to the authors.

---

> > > ### Author Response · Authors · 2026-04-06
> > >
> > > Thank you. We address your remaining concerns below.
> > >
> > > ## On the Method Being "A Sequence of LLM Prompting"
> > >
> > > Our method operates through LLM prompting — but so does every inference-time self-correction method: Self-Refine prompts the model to critique and regenerate, CoVe derives verification questions from the chain-of-thought. If relying on prompting limits novelty, this applies equally to the entire subfield.
> > >
> > > What distinguishes Thought-ICS is the pipeline design. We decompose self-correction into verification, localization, and resampling as independently studied components — something no other paper does at our scale. This enabled the discovery of specificity collapse: that verification, not correction capability, is the bottleneck. Our system localizes the first erroneous thought and resamples from the verified-correct prefix, rather than regenerating from holistic critique. Resampling from a clean prefix yields 2–4x higher edit accuracy than from an erroneous prefix (Fig. 5), and produces a 77.5% answer-change rate vs 58.9% for token-level approaches. The self-refinement approaches in the literature, whether inference-time or training-time, reduce to asking the model to evaluate its reasoning and generate a correction. We provide a principled framework for the correction step that future work can use to produce consistently better corrections, irrespective of whether verification is provided or self-guided.
> > >
> > > ## Our Key Finding is Novel
> > >
> > > Huang et al. (2024) and Stechly et al. (2024) establish that LLMs struggle to self-correct without external signals. Tyen et al. (2024) find that LLMs cannot self-localize errors — and train a separate localizer to do so. We show that with thought-level structure, models across scales (3B–120B) can self-localize errors *without any further training*, enabling effective correction (20–40% lift under oracle verification). No other paper establishes this point. This overturns a key prior result and has implications beyond test-time methods — it implies models can self-assign variable credit to reasoning prefixes for credit assignment during training. Our breadth (8 models, 6 benchmarks) reveals how localization, correction, and verification each scale with model size.
> > >
> > > ## On Baselines
> > >
> > > **In-regime comparisons.** Within our setting (single model, inference-time, no auxiliary training, no external environment), by the taxonomy of Kamoi et al. (TACL 2024; arxiv:2406.01297), Self-Refine and CoVe remain the only truly comparable methods.
> > >
> > > **Out-of-regime comparisons.** Methods requiring training of the reasoner or an external verifier are out of scope for our paper. Two notable recent examples:
> > >
> > > - StepCo (arxiv:2410.12934): trains a Process-Supervised Verifier for step-level verification and localization; the base LLM never self-verifies or self-localizes, and their evaluation includes no LLM self-localization baseline.
> > > - SCoRe (arxiv:2409.12917): multi-turn online RL on self-generated correction traces; at inference the trained model just generates two attempts.
> > >
> > > For the reviewer's reference, we ran a quick StepCo test on Qwen-7B and Qwen-14B using Math-Shepherd, one of the off-the-shelf PRMs evaluated in their paper (StepCo's custom PSV is reported as +0.9 avg stronger).
> > >
> > > **StepCo head-to-head, Qwen-7B and Qwen-14B**
> > >
> > > *Qwen-7B*
> > >
> > > | Method | AMC23 | AIME | MATH-L5 | CSQA | GPQA | MathQA |
> > > |---|---:|---:|---:|---:|---:|---:|
> > > | Unstructured CoT | 37.5 | **16.0** | 35.0 | 71.0 | 26.0 | 69.0 |
> > > | StepCo | **45.0** | 12.0 | 28.0 | 71.0 | 23.0 | 62.0 |
> > > | Thought-ICS-A (ours) | 40.0 | **16.0** | **40.0** | **85.0** | **28.0** | **71.0** |
> > >
> > > ---
> > >
> > > *Qwen-14B*
> > >
> > > | Method | AMC23 | AIME | MATH-L5 | CSQA | GPQA | MathQA |
> > > |---|---:|---:|---:|---:|---:|---:|
> > > | Unstructured CoT | 45.0 | 18.0 | 40.0 | 84.0 | 35.0 | **84.0** |
> > > | StepCo | **62.5** | 15.0 | 38.0 | 79.0 | 36.0 | 77.0 |
> > > | Thought-ICS-A (ours) | 50.0 | **20.0** | **43.0** | **90.0** | **38.0** | 83.0 |
> > >
> > > Training a PRM and using it in StepCo's pipeline does not consistently beat our structured self-localization without any training. Across all 12 cells (2 models × 6 benchmarks), Thought-ICS-A wins 10 head-to-heads against StepCo, losing only on AMC23. Table 5 of the appendix shows Thought-ICS-A beats contemporary baselines (Self-Refine, CoVe) on 14B–120B models in 26/30 cases — the scale at which verification, localization, and correction become consistent within our framework. Under oracle verification (Table 4, p.25), Thought-ICS reaches 77.5/32.0/58.0/94.0/66.0 on Qwen-7B (AMC23/AIME/MATH-L5/CSQA/GPQA), indicating the correction mechanism is strong and the autonomous gap reflects poor verification.
> > > In light of the above — overturning prior self-localization results, the principled correction framework, and baseline comparisons with expanded scope — we respectfully ask the reviewer to reconsider increasing their score.

---

### Official Review · Reviewer_QXwV · 2026-03-13

**Soundness:** 3
**Presentation:** 3
**Significance:** 3
**Originality:** 3
**Overall Recommendation:** 5
**Confidence:** 3

**Summary:**

This paper proposes the Thought-ICS framework, which aims to iteratively generate reasoning one complete thought at a time. It reframes LLM inference as reasoning step generation, and structures it as a MDP. This allows for self-correction to outperform CoT baselines through this cleaner generation method.

**Compliance With Llm Reviewing Policy:**

Affirmed.

**Final Justification:**

The rebuttal addressed most of my concerns, and I believe that the paper has strong impact so I will change my evaluation score to 5.

**Key Questions For Authors:**

1. Is there a significant change in token efficiency or other overhead with Thought-ICS vs existing frameworks?
2. The results seem to differ drastically over different tasks (also when comparing with baseline methods, certain tasks Thought-ICS is not the best)? Why might this be the case?

**Limitations:**

Yes

**Strengths And Weaknesses:**

**Soundness**

Strengths:
- Evaluation is extensive across many different models and framework is well-defined
- Demonstrates improvements over previous self-correction baselines with CoT even in fully autonomous setting

Weaknesses:
- Oracle verification seems to make the problem a lot more simple and undermines the underlying story about self-correction

**Presentation**

Strengths:
- The submission is well structured and easy to follow

Weaknesses:
- Some analysis on computational cost may improve the practical evaluation
- Oracle components could be mentioned more clearly in the method section rather than experiments as it seems fairly important to the results

**Significance**

Strengths:
- The paper has important findings relating to self-localization using this thought-based breakdown which can be applied in future inference systems
- Contributes a solid framework to evaluate self-correction by reframing the LLM reasoning process

Weaknesses:
- Focuses mostly on tasks that have concrete answers, doesn't address general reasoning processes
- Improvements depend heavily on oracle verification

**Originality**

- Introduces a novel combination of thought-level breakdown and self-correction
- Focuses on localized correction

---

> ### Author Rebuttal · Authors · 2026-03-31
>
> We thank the reviewer for their positive assessment and constructive feedback.
>
> **W1: Oracle verification simplifies the problem and undermines the self-correction story.**
>
> Oracle verification does not undermine our findings—it is a deliberate experimental
> design choice that allows us to study the model's corrective ability (localization +
> resampling) isolated from verification failures. This decomposition is a core contribution:
> we show that when verification is reliable, models can successfully localize errors and fix
> them. We then separately analyze *why* autonomous self-verification fails (Section 6),
> identifying specificity collapse as the specific bottleneck, and take tangible steps toward
> mitigating it. Self-verification of difficult reasoning problems is an unsolved challenge
> for language models broadly—our paper provides a structured empirical analysis of where
> and how it breaks down, rather than treating it as a black box.
>
> **W2: Missing computational cost analysis.**
>
> See Q1 below.
>
> **W3: Oracle components should be in method section, not just experiments.**
>
> We do discuss both oracle and self-verified settings in the method section (Section 4,
> p.3). We will review the presentation to ensure this distinction is sufficiently clear.
>
> **W4: Limited to tasks with concrete answers; improvements depend heavily on oracle verification.**
>
> We focus on verifiable tasks to enable rigorous measurement, but the framework generalizes
> to any domain with a verification signal (e.g., code execution). Regarding oracle
> dependence: under fully autonomous self-verification, Thought-ICS-A still outperforms all
> baselines (Figure 8), using the fewest correction tokens ($1{,}133$) at the highest accuracy
> ($62.9%$). The oracle experiments isolate correction capability from verification—they do
> not mean the method requires an oracle to be useful.
>
> **Q1: Significant change in token efficiency or overhead with Thought-ICS vs existing frameworks?**
>
> Yes. Under oracle verification, Thought-ICS uses $1.16\times$ more tokens than Token-ICS
> but is $1.2\times$ more token-efficient per accuracy point ($30.3$ vs $36.4$
> tokens/acc-point). In the autonomous setting, Thought-ICS-A is the most efficient method:
> $1{,}133$ correction tokens at $62.9%$ accuracy ($18.0$ tokens/acc-point), compared to
> Self-Refine ($1{,}415$ tokens, $46.6%$, $30.4$) and CoVe ($2{,}592$ tokens, $46.3%$,
> $56.0$).
>
> **Q2: Results differ drastically over different tasks; Thought-ICS is not always best. Why?**
>
> We test 8 models spanning different parameter sizes and training vintages to get
> comprehensive coverage. Under oracle verification, Thought-ICS consistently wins across all
> models and datasets. In the autonomous setting, variation comes from two sources: (1) model
> capability—larger models (14B+) are better able to utilize the self-correction framework,
> and Thought-ICS consistently beats baselines for these models (Table 5, p.27 for detailed
> breakdown); and (2) specificity collapse, the failure mode we identify in Section 6, where
> the verifier incorrectly rejects correct answers. The prevalence of this failure mode varies across models and tasks, which we believe
> contributes to the variation in autonomous effectiveness.
>
> We thank the reviewer for recognizing the framework's contributions and hope the token
> efficiency analysis and task variability discussion strengthen the paper. We will
> incorporate the suggested presentation changes in the revision and would be grateful if the
> reviewer would consider increasing their score.

---

> > ### Author Rebuttal · Reviewer_QXwV · 2026-04-02
> >
> > Thanks for the responses, and my concerns have been mostly addressed.

---

> > > ### Author Response · Authors · 2026-04-07
> > >
> > > Thank you for the comments and efforts.

---

### Official Review · Reviewer_DqUW · 2026-03-13

**Soundness:** 3
**Presentation:** 2
**Significance:** 2
**Originality:** 2
**Overall Recommendation:** 3
**Confidence:** 4

**Summary:**

This work presents a self-localization and self-correction framework for LLMs. The main idea is to prompt the model to generate the reasoning steps step-by-step. This work also proposes verification, localization and resampling strategies based on that. The empirical study is conducted on 8 models with different sizes, and on 6 reasoning benchmarks. The results show that structured reasoning steps makes it easier to self-correct and self-localize. However, for a fully autonomous system, the recall-specificity tradeoff is hard to break. Finally, the proposed method is shown to outperform 2 methods as an inference-time correction method.

**Compliance With Llm Reviewing Policy:**

Affirmed.

**Final Justification:**

The rebuttal addressed some of my concerns. However, I'm still keeping my original score, as although I think this paper studies an interesting angle, the impact is a bit limited, and there are some questions about the evaluation.

**Key Questions For Authors:**

1. Does token-ICS use the same system instruction?
2. Have you done any human study on the oracle localization accuracy? I also wonder if the oracle models are more noisy for token-level localization, which might be a confounding factor for some experiments in this work.
3. How do you make sure the resampling is different? What do you do if you always get similar samples?

**Limitations:**

yes

**Strengths And Weaknesses:**

Strengths
1. The empirical study is detailed and thorough. I especially like the angle of recall-specificity tradeoff, which provides a more comprehensive view of evaluation.
2. Self-improvements of LLM is an important and interesting topic that has great potential in practice.

Weaknesses
1. The proposed method depends on prompting the model to generated reasoning steps in a particular structure. I’m not sure how that will influence the model’s performance (compared to free generation). This is also not verified in the empirical results in this work.
2. For inference time methods, an important part of the evaluation is the computation cost, but this is not mentioned in the paper.
3. A few important details are not clear. See questions below.

---

> ### Author Rebuttal · Authors · 2026-03-31
>
> We thank the reviewer for their thorough evaluation. We address each concern below.
>
> **W1: Structured generation vs free generation performance.**
>
> The structured generation simply wraps each reasoning step in `<thought>` tags—this
> is analogous to the demarcation tokens frontier LLMs already use in standard generation
> (e.g., paragraph breaks, numbered steps). Table 3 (Appendix D) directly compares structured vs unstructured generation
> accuracy: structured formatting generally does not degrade generation performance, and larger models
> are better able to leverage the structure.
>
> **W2: Computation cost not discussed.**
>
> We conducted a compute analysis measuring new output tokens per problem during the
> correction phase (excluding the shared KV-cached initial CoT).
>
> **L2 (oracle verification), Thought-ICS vs Token-ICS.** Under oracle verification, the
> system iterates until the answer is correct or max iterations is reached, so incorrect
> problems generate more correction tokens.
>
> | Method | Corr. Tokens | Acc.% | Tok/Acc-pt |
> |--------|-------|-------|-------|
> | Token-ICS | $1{,}750$ | $48.1$ | $36.4$ |
> | Thought-ICS | $2{,}039$ | $67.2$ | $30.3$ |
>
> Thought-ICS uses $1.16\times$ more tokens but achieves $+19.1$ pp accuracy, making it
> $1.2\times$ more token-efficient per accuracy point. For larger models (Qwen-32B, OSS-120B),
> Thought-ICS uses *fewer* tokens due to faster convergence.
>
> **Autonomous (L3), all methods.** Here the model self-verifies and stops when it believes
> its answer is correct, so fewer correction tokens are generated overall (but accuracy is
> lower due to imperfect self-verification).
>
> | Method | Corr. Tokens | Acc.% | Tok/Acc-pt |
> |--------|-------|-------|-------|
> | CoVe | $2{,}592$ | $46.3$ | $56.0$ |
> | Self-Refine | $1{,}415$ | $46.6$ | $30.4$ |
> | Thought-ICS-A | $1{,}133$ | $62.9$ | $18.0$ |
>
> Thought-ICS-A is the most token-efficient method in the autonomous setting, using the
> fewest correction tokens while achieving the highest accuracy.
>
> **Q1: Does Token-ICS use the same system instruction?**
>
> Neither method uses a system instruction. Both use the same base model with the same
> generation prompt ("Solve the following math problem step by step..."). The only
> difference is that Thought-ICS asks the model to wrap each step in `<thought>` tags,
> while Token-ICS uses step-by-step CoT without explicit demarcation tokens. The localization and
> resampling prompts differ by necessity (step-level vs character-level). All prompts are
> detailed in Appendix C (Thought-ICS) and Appendix E (baselines including Token-ICS).
>
> **Q2: Human study on oracle localization accuracy? Oracle noise for token-level localization as confounding factor?**
>
> We did not conduct a human study, but we measure oracle localization consistency using three
> frontier models (Sonnet-3.7, GPT-4.1, GPT-5-mini) and report agreement rates (Appendix G).
> For Token-ICS, we use a relaxed 10% tolerance threshold to account for the inherent
> ambiguity of continuous localization. These are inherent limitations of the unstructured format—without discrete thought
> boundaries, it is more difficult for models to clearly identify erroneous steps, which is
> exactly what existing refinement approaches ask the model to implicitly do. Providing
> structure makes localization more reliable. See Appendix G (p.23) for details.
>
> **Q3: How do you ensure resampling diversity? What if samples are always similar?**
>
> We use temperature sampling ($t{=}0.7$) to resample on-policy corrections conditioned on
> the verified-correct prefix. Rather than including the previous erroneous reasoning trace
> in context—which we find is not more performant for smaller models, likely because they
> struggle to reliably utilize that information—we simply condition generation on the shared
> prefix the model has verified to be accurate. Backtracking to the error step controls which
> prefix the model resamples from, enabling diverse continuations without requiring the model
> to reason about its prior mistakes.
>
> We conducted a post-hoc variability analysis (1,631 Thought-ICS problems, 1,064 Token-ICS,
> each with $>$1 correction iteration). Across consecutive iterations, Token-ICS produces the
> *same answer* 44.6% of the time versus 30.6% for Thought-ICS. 77.5% of Thought-ICS
> corrections change the answer from the initial attempt, compared to 58.9% for Token-ICS.
> The gap widens for larger models (OSS-20B/120B: 76--79% repetition for Token-ICS vs
> 50--55% for Thought-ICS). These results show that step-level backtracking enables
> meaningfully more diverse resampled corrections across all model scales.
>
> We hope the new compute analysis, variability results, and clarifications on oracle
> localization address the reviewer's concerns. We would be grateful if the reviewer would
> consider increasing their score in light of these additions.

---

> > ### Author Rebuttal · Reviewer_DqUW · 2026-04-05
> >
> > Thank you for the response. I still have concerns about the evaluation of the proposed method as inference time methods. When compared with other baselines, the computation cost (ideally, the entire tradeoff curve) should be reported. I also feel the 51% unanimous agreement is not super high, so there should have a discussion on how reliable and convincing the reported metrics are. Hence, I keep my score unchanged for now.

---

> > > ### Author Response · Authors · 2026-04-06
> > >
> > > We thank the reviewer for the follow-up. We address both remaining concerns below.
> > >
> > >
> > > ## Computation Cost and Tradeoff Curves
> > >
> > >
> > > We appreciate the suggestion to report the full tradeoff curve. We do have iteration-by-iteration performance data for all methods and would be happy to include full tradeoff curves in the camera-ready; however, the rebuttal format does not permit us to add new figures.
> > >
> > >
> > > That said, the computational analysis we provided in our previous response is representative of the full tradeoff. The correction tokens, accuracy, and tokens-per-accuracy-point we reported are *terminal* values that reflect the cumulative cost and performance across all iterations. As shown, Thought-ICS-A achieves the highest accuracy while using the fewest correction tokens — 1.7x more token-efficient than Self-Refine and 3.1x more than CoVe per accuracy point. Under oracle verification, Thought-ICS uses only 1.16x more tokens than Token-ICS but achieves +19.1 pp higher accuracy, and for larger models (Qwen-32B, OSS-120B) actually uses *fewer* tokens due to faster convergence. These terminal values capture the essential tradeoff: more accuracy for less (or comparable) compute.
> > >
> > >
> > > ## On the 51% Unanimous Agreement and Metric Reliability
> > >
> > >
> > > We understand the concern, but we believe the 51% figure is being interpreted out of context. The full agreement profile from Appendix G (p. 23) is:
> > >
> > >
> > > - **51%** unanimous agreement (all three frontier oracles agree on the exact same step)
> > > - **74%** within ±1 step
> > > - **85%** within ±2 steps
> > >
> > >
> > > In reasoning traces that are 10–20 thoughts long, ±1–2 steps of disagreement between independent frontier-model oracles is reasonable — these are judgment calls about where exactly an error first manifests, and minor variation is expected. The 85% within ±2 steps represents strong agreement for this task.
> > >
> > >
> > > More importantly, **the reliability of our findings does not rest solely on unanimous oracle agreement.** We specifically controlled for this in our analysis:
> > >
> > >
> > > 1. **Controlled analysis on agreed-upon localizations.** We report localization quality metrics on the subset of cases where oracles agreed *and* produced valid localizations — ensuring the cleanest possible comparison. The findings from this controlled subset are **consistent with the findings from the full set of all localization attempts**, meaning our conclusions are robust to the inclusion of noisier cases.
> > >
> > >
> > > 2. **The key finding holds regardless of oracle reliability.** Irrespective of how reliable one considers the oracle determination to be, there is a clear and consistent difference between the edit accuracies of *clean prefixes* (derived from accurate localizations) and *erroneous prefixes* (derived from inaccurate/later localizations that missed the first erroneous thought). As we report in Section 4 (Fig. 5), resampling from a clean prefix yields a **2–4x improvement** in edit accuracy over resampling from an erroneous prefix — a gap observed consistently across models and datasets. The signal is strong enough to support our conclusions even without perfect oracle agreement.
> > >
> > >
> > > 3. **Unstructured methods face inherently worse localization reliability.** Any difficulty in obtaining clean localization analysis in non-thought-MDP-based methods is inherent to the lack of structure in their responses. Without clearly marked thought boundaries — when reasoning is generated as a continuous stream — localization is fundamentally more ambiguous, whether done implicitly by the model or explicitly by an oracle. This is precisely the point of our work: thought-level structure enables more reliable localization and thus more effective self-correction, rather than relying on the model to handle these things implicitly within its context window as prior methods do.
> > >
> > >
> > > We hope these clarifications address the reviewer's remaining concerns. The computational analysis demonstrates favorable cost-performance tradeoffs, and the oracle agreement — when viewed in full (85% within ±2 steps) and in conjunction with the controlled analyses we performed — provides a reliable basis for our findings. As such, we kindly request that the reviewer consider updating their score in light of these clarifications and the detailed discussions that already present in our paper/appendix addressing their points.

---

### Decision · Program_Chairs · 2026-04-30

**Decision:**

Accept (regular)

**Comment:**

The paper proposed a Thought-ICS (including verification, localization and resampling strategies), which is actually a prompting based method on LLMs for automated self-correction. The main idea is to prompt the model to generate the reasoning steps step-by-step. The results show that the proposed structured reasoning steps makes it easier to self-correct and self-localize than token-level ICS. According to all reviews, the main concerns are as follows.

1. The proposed method depends on prompting the model to generated reasoning steps in a particular structure. In this way, the contribution is limited although the finding is novel. And the computation cost in the inference time should be discussed. The authors have provided some results in the rebuttals, it is better to add them to the revised version.
2. More related work should be added.